# Multiresolution Equivariant Graph Variational Autoencoder

## Abstract

In this paper, we propose *Multiresolution Equivariant Graph Variational Autoencoders* (MGVAE), the first hierarchical generative model to learn and generate graphs in a multiresolution and equivariant manner. At each resolution level, MG-VAE employs higher order message passing to encode the graph while learning to partition it into mutually exclusive clusters and coarsening into a lower resolution that eventually creates a hierarchy of latent distributions. MGVAE then constructs a hierarchical generative model to variationally decode into a hierarchy of coarsened graphs. Importantly, our proposed framework is end-to-end permutation equivariant with respect to node ordering. MGVAE achieves competitive results with several generative tasks including general graph generation, molecular generation, unsupervised molecular representation learning to predict molecular properties, link prediction on citation graphs, and graph-based image generation.

## 1 Introduction

Understanding graphs in a multiscale and multiresolution perspective is essential for capturing the structure of molecules, social networks, or the World Wide Web. Graph neural networks (GNNs) utilizing various ways of generalizing the concept of convolution to graphs (Scarselli et al., 2009) (Niepert et al., 2016b) (Li et al., 2016) have been widely applied to many learning tasks, including modeling physical systems (Battaglia et al., 2016), finding molecular representations to estimate quantum chemical computation (Duvenaud et al., 2015) (Kearnes et al., 2016) (Gilmer et al., 2017b) (Hy et al., 2018), and protein interface prediction (Fout et al., 2017). One of the most popular types of GNNs is message passing neural nets (MPNNs) that are constructed based on the message passing scheme in which each node propagates and aggregates information, encoded by vectorized messages, to and from its local neighborhood. While this framework has been immensely successful in many applications, it lacks the ability to capture the multiscale and multiresolution structures that are present in complex graphs (Rustamov & Guibas, 2013) (Chen et al., 2014) (Cheng et al., 2015) (Xu et al., 2019).

Ying et al. (2018) proposed a multiresolution graph neural network that employs a differential pooling operator to coarsen the graph. While this approach is effective in some settings, it is based on *soft* assigment matrices, which means that (a) the sparsity of the graph is quickly lost in higher layers (b) the algorithm isn't able to learn an actual hard clustering of the vertices. The latter is important in applications such as learning molecular graphs, where clusters should ideally be interpretable as concrete subunits of the graphs, e.g., functional groups.

In contrast, in this paper we propose an arhictecture called *Multiresolution Graph Network (MGN)*, and its generative cousin, *Multiresolution Graph Variational Autoencoder* (MGVAE), which explicitly learn a multilevel hard clustering of the vertices, leading to a true multiresolution hierarchy. In the decoding stage, to "uncoarsen" the graph, MGVAE needs to generate local adjacency matrices, which is inherently a second order task with respect to the action of permutations on the vertices, hence MGVAE needs to leverage the recently developed framework of higher order permutation equivariant message passing networks (Hy et al., 2018; Maron et al., 2019b).

Learning to generate graphs with deep generative models is a difficult problem because graphs are combinatorial objects that typically have high order correlations between their discrete substructures (subgraphs) (You et al., 2018a) (Li et al., 2018) (Liao et al., 2019) (Liu et al., 2019) (Dai et al., 2020). Graph-based molecular generation (Gmez-Bombarelli et al., 2018) (Simonovsky &

Komodakis, 2018) (Cao & Kipf, 2018) (Jin et al., 2018) (Thiede et al., 2020) involves further challenges, including correctly recognizing chemical substructures, and importantly, ensuring that the generated molecular graphs are chemically valid. MGN allows us to extend the existing model of variational autoencoders (VAEs) with a hierarchy of latent distributions that can stochastically generate a graph in multiple resolution levels. Our experiments show that having a flexible clustering procedure from MGN enables MGVAE to detect, reconstruct and finally generate important graph substructures, especially chemical functional groups.

## 2 RELATED WORK

There have been significant advances in understanding the invariance and equivariance properties of neural networks in general (Cohen & Welling, 2016a) (Cohen & Welling, 2016b), of graph neural networks (Maron et al., 2019b), of neural networks learning on sets (Zaheer et al., 2017) (Serviansky et al., 2020) (Maron et al., 2020), along with their expressive power on graphs (Maron et al., 2019c) (Maron et al., 2019a). Our work is in line with group equivariant networks operating on graphs and sets. Multiscale, multilevel, multiresolution and coarse-grained techniques have been widely applied to graphs and discrete domains such as diffusion wavelets (Coifman & Maggioni, 2006); spectral wavelets on graphs (Hammond et al., 2011); finding graph wavelets based on partitioning/clustering (Rustamov & Guibas, 2013); graph clustering and finding balanced cuts on large graphs (Dhillon et al., 2005) (Dhillon et al., 2007) (Chiang et al., 2012) (Si et al., 2014); and link prediction on social networks (Shin et al., 2012). Prior to our work, some authors such as (Zhou et al., 2019) proposed a multiscale generative model on graphs using GAN (Goodfellow et al., 2014), but the hierarchical structure was built by heuristics algorithm, not learnable and not flexible to new data that is also an existing limitation of the field. In general, our work exploits the powerful group equivariant networks to encode a graph and to learn to form balanced partitions via back-propagation in a data-driven manner without using any heuristics as in the existing works.

In the field of deep generative models, it is generally recognized that introducing a hierarchy of latents and adding stochasticity among latents leads to more powerful models capable of learning more complicated distributions (Blei et al., 2003) (Ranganath et al., 2016) (Ingraham & Marks, 2017) (Klushyn et al., 2019) (Wu et al., 2020) (Vahdat & Kautz, 2020). Our work combines the hierarchical variational autoencoder with learning to construct the hierarchy that results into a generative model able to generate graphs at many resolution levels.

## 3 MULTIRESOLUTION GRAPH NETWORK

### 3.1 CONSTRUCTION

An undirected weighted graph $\mathcal{G} = (\mathcal{V}, \mathcal{E}, \mathcal{A}, \mathcal{F}_v, \mathcal{F}_e)$ with node set $\mathcal{V}$ and edge set $\mathcal{E}$ is represented by an adjacency matrix $\mathcal{A} \in \mathbb{N}^{|\mathcal{V}| \times |\mathcal{V}|}$, where $\mathcal{A}_{ij} > 0$ implies an edge between node $v_i$ and $v_j$ with weight $\mathcal{A}_{ij}$ (e.g., $\mathcal{A}_{ij} \in \{0, 1\}$ in the case of unweighted graph); while node features are represented by a matrix $\mathcal{F}_v \in \mathbb{R}^{|\mathcal{V}| \times d_v}$, and edge features are represented by a tensor $\mathcal{F}_e \in \mathbb{R}^{|\mathcal{V}| \times |\mathcal{V}| \times d_e}$. The second-order tensor representation of edge features is necessary for our higher-order message passing networks described in the next section. Indeed, $\mathcal{F}_v$ can be encoded in the diagonal of $\mathcal{F}_e$.

**Definition 1.** *A K-cluster partition of graph $\mathcal{G}$ is a partition of the set of nodes $\mathcal{V}$ into $K$ mutually exclusive clusters $\mathcal{V}_1, .., \mathcal{V}_K$. Each cluster corresponds to an induced subgraph $\mathcal{G}_k = \mathcal{G}[\mathcal{V}_k]$.*

**Definition 2.** *A coarsening of $\mathcal{G}$ is a graph $\tilde{\mathcal{G}}$ of $K$ nodes defined by a K-cluster partition in which node $\tilde{v}_k$ of $\tilde{\mathcal{G}}$ corresponds to the induced subgraph $\mathcal{G}_k$. The weighted adjacency matrix $\tilde{\mathcal{A}} \in \mathbb{N}^{K \times K}$ of $\tilde{\mathcal{G}}$ is*

$$\tilde{\mathcal{A}}_{kk'} = \begin{cases} \frac{1}{2} \sum_{v_i, v_j \in \mathcal{V}_k} \mathcal{A}_{ij}, & \text{if } k = k', \\ \sum_{v_i \in \mathcal{V}_k, v_j \in \mathcal{V}_{k'}} \mathcal{A}_{ij}, & \text{if } k \neq k', \end{cases}$$

*where the diagonal of $\tilde{\mathcal{A}}$ denotes the number of edges inside each cluster, while the off-diagonal denotes the number of edges between two clusters.*

Fig. 3.1 shows an example of Defs. 1 and 2: a 3-cluster partition of the Aspirin $C_9H_8O_4$ molecular graph and its coarsening graph. Def. 3 defines the multiresolution of graph $\mathcal{G}$ in a bottom-up manner

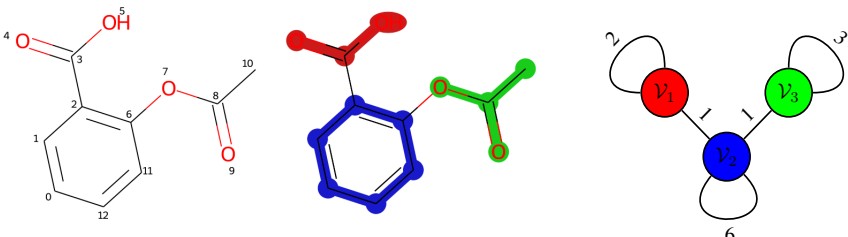

Figure 1: Aspirin $C_9H_8O_4$, its 3-cluster partition and the corresponding coarsen graph

in which the bottom level is the highest resolution (e.g., $\mathcal{G}$ itself) while the top level is the lowest resolution (e.g., $\mathcal{G}$ is coarsened into a single node).

**Definition 3.** *An $L$-level coarsening of a graph $\mathcal{G}$ is a series of $L$ graphs $\mathcal{G}^{(1)}, .., \mathcal{G}^{(L)}$ in which*

1. *$\mathcal{G}^{(L)}$ is $\mathcal{G}$ itself.*

2. *For $1 \leq \ell \leq L - 1$, $\mathcal{G}^{(\ell)}$ is a coarsening graph of $\mathcal{G}^{(\ell+1)}$ as defined in Def. 2. The number of nodes in $\mathcal{G}^{(\ell)}$ is equal to the number of clusters in $\mathcal{G}^{(\ell+1)}$.*

3. *The top level coarsening $\mathcal{G}^{(1)}$ is a graph consisting of a single node, and the corresponding adjacency matrix $\mathcal{A}^{(1)} \in \mathbb{N}^{1 \times 1}$.*

**Definition 4.** *An $L$-level Multiresolution Graph Network (MGN) of a graph $\mathcal{G}$ consists of $L - 1$ tuples of five network components $\{(\boldsymbol{c}^{(\ell)}, \boldsymbol{e}_{local}^{(\ell)}, \boldsymbol{d}_{local}^{(\ell)}, \boldsymbol{d}_{global}^{(\ell)}, \boldsymbol{p}^{(\ell)})\}_{\ell=2}^{L}$. The $\ell$-th tuple encodes $\mathcal{G}^{(\ell)}$ and transforms it into a lower resolution graph $\mathcal{G}^{(\ell-1)}$ in the higher level. Each of these network components has a separate set of learnable parameters $(\boldsymbol{\theta}_1^{(\ell)}, \boldsymbol{\theta}_2^{(\ell)}, \boldsymbol{\theta}_3^{(\ell)}, \boldsymbol{\theta}_4^{(\ell)}, \boldsymbol{\theta}_5^{(\ell)})$. For simplicity, we collectively denote the learnable parameters as $\boldsymbol{\theta}$, and drop the superscript. The network components are defined as follows:*

1. *Clustering procedure $\boldsymbol{c}(\mathcal{G}^{(\ell)}; \boldsymbol{\theta})$, which partitions graph $\mathcal{G}^{(\ell)}$ into $K$ clusters $\mathcal{V}_1^{(\ell)}, \ldots, \mathcal{V}_K^{(\ell)}$. Each cluster is an induced subgraph $\mathcal{G}_k^{(\ell)}$ of $\mathcal{G}^{(\ell)}$ with adjacency matrix $\mathcal{A}_k^{(\ell)}$.*

2. *Local encoder $\boldsymbol{e}_{local}(\mathcal{G}_k^{(\ell)}; \boldsymbol{\theta})$, which is a permutation equivariant (see Defs. 5, 6) graph neural network that takes as input the subgraph $\mathcal{G}_k^{(\ell)}$, and outputs a set of node latents $\mathcal{Z}_k^{(\ell)}$ represented as a matrix of size $|\mathcal{V}_k^{(\ell)}| \times d_z$.*

3. *Local decoder $\boldsymbol{d}_{local}(\mathcal{Z}_k^{(\ell)}; \boldsymbol{\theta})$, which is a permutation equivariant neural network that tries to reconstruct the subgraph adjacency matrix $\mathcal{A}_k^{(\ell)}$ for each cluster from the local encoder's output latents.*

4. *(Optional) Global decoder $\boldsymbol{d}_{global}(\mathcal{Z}^{(\ell)}; \boldsymbol{\theta})$, which is a permutation equivariant neural network that reconstructs the full adjacency matrix $\mathcal{A}^{(\ell)}$ from all the node latents of $K$ clusters $\mathcal{Z}^{(\ell)} = \bigoplus_k \mathcal{Z}_k^{(\ell)}$ represented as a matrix of size $|\mathcal{V}^{(\ell)}| \times d_z$.*

5. *Pooling network $\boldsymbol{p}(\mathcal{Z}_k^{(\ell)}; \boldsymbol{\theta})$, which is a permutation invariant (see Defs. 5, 6) neural network that takes the set of node latents $\mathcal{Z}_k^{(\ell)}$ and outputs a single cluster latent $\tilde{z}_k^{(\ell)} \in d_z$. The coarsening graph $\mathcal{G}^{(\ell-1)}$ has adjacency matrix $\mathcal{A}^{(\ell-1)}$ built as in Def. 2, and the corresponding node features $\mathcal{Z}^{(\ell-1)} = \bigoplus_k \tilde{z}_k^{(\ell)}$ represented as a matrix of size $K \times d_z$.*

Algorithmically, MGN works in a bottom-up manner as a tree-like hierarchy starting from the highest resolution graph $\mathcal{G}^{(L)}$, going to the lowest resolution $\mathcal{G}^{(1)}$ (see Fig. 3.1). Iteratively, at $\ell$-th level, MGN partitions the current graph into $K$ clusters by running the clustering procedure $\boldsymbol{c}^{(\ell)}$. Then, the local encoder $\boldsymbol{e}_{local}^{(\ell)}$ and local decoder $\boldsymbol{d}_{global}^{(\ell)}$ operate on each of the $K$ subgraphs separately, and can be executed in parallel. This encoder/decoder pair is responsible for capturing the local structures. Finally, the pooling network $\boldsymbol{p}^{(\ell)}$ shrinks each cluster into a single node of the next level.

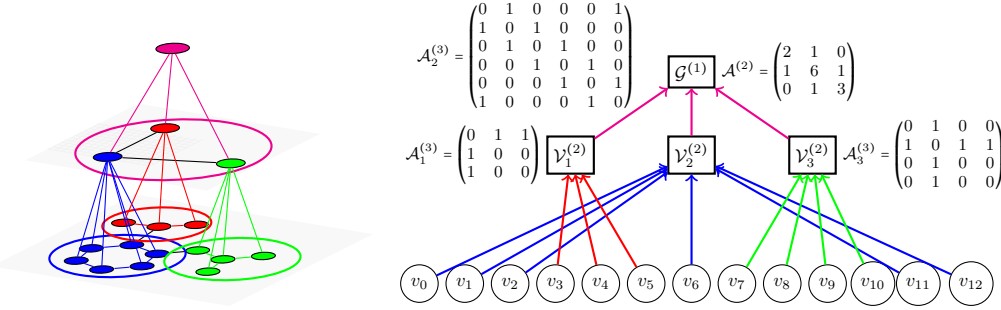

Figure 2: Hierarchy of 3-level Multiresolution Graph Network on Aspirin molecular graph

Optionally, the global decoder $d_{\text{global}}^{(\ell)}$ makes sure that the whole set of node latents $\mathcal{Z}^{(\ell)}$ is able to capture the inter-connection between clusters.

In terms of time and space complexity, MGN is more efficient than existing methods in the field. The cost of global decoding the highest resolution graph is proportional to $|\mathcal{V}|^2$. For example, while the encoder can exploit the sparsity of the graph and has complexity $\mathcal{O}(|\mathcal{E}|)$, a simple dot-product decoder $d_{\text{global}}(\mathcal{Z}) = \text{sigmoid}(\mathcal{Z}\mathcal{Z}^T)$ has both time and space complexity of $\mathcal{O}(|\mathcal{V}|^2)$ which is infeasible for large graphs. In contrast, the cost of running $K$ local dot-product decoders is $\mathcal{O}(|\mathcal{V}|^2/K)$, which is approximately $K$ times more efficient.

## 3.2 HIGHER ORDER MESSAGE PASSING

In this paper we consider permutation symmetry, i.e., symmetry to the action of the symmetric group, $\mathbb{S}_n$. An element $\sigma \in \mathbb{S}_n$ is a permutation of order $n$, or a bijective map from $\{1, \dots, n\}$ to $\{1, \dots, n\}$. The action of $\mathbb{S}_n$ on an adjacency matrix $\mathcal{A} \in \mathbb{R}^{n \times n}$ and on a latent matrix $\mathcal{Z} \in \mathbb{R}^{n \times d_z}$ are

$$[\sigma \cdot \mathcal{A}]_{i_1,i_2} = \mathcal{A}_{\sigma^{-1}(i_1),\sigma^{-1}(i_2)}, \qquad [\sigma \cdot \mathcal{Z}]_{i,j} = \mathcal{Z}_{\sigma^{-1}(i),j}, \qquad \sigma \in \mathbb{S}_n.$$

Here, the adjacency matrix $\mathcal{A}$ is a second order tensor with a single feature channel, while the latent matrix $\mathcal{Z}$ is a first order tensor with $d_z$ feature channels. In general, the action of $\mathbb{S}_n$ on a $k$-th order tensor $\mathcal{X} \in \mathbb{R}^{n^k \times d}$ (the last index denotes the feature channels) is defined similarly as:

$$[\sigma \cdot \mathcal{X}]_{i_1,..,i_k,j} = \mathcal{X}_{\sigma^{-1}(i_1),..,\sigma^{-1}(i_k),j}, \qquad \sigma \in \mathbb{S}_n.$$

Network components of MGN (as defined in Sec. 3.1) at each resolution level must be either *equivariant*, or *invariant* with respect to the permutation action on the node order of $\mathcal{G}^{(\ell)}$. Formally, we define these properties in Def. 5.

**Definition 5.** *An $\mathbb{S}_n$-equivariant (or permutation equivariant) function is a function $f\colon \mathbb{R}^{n^k \times d} \to \mathbb{R}^{n^{k'} \times d'}$ that satisfies $f(\sigma \cdot \mathcal{X}) = \sigma \cdot f(\mathcal{X})$ for all $\sigma \in \mathbb{S}_n$ and $\mathcal{X} \in \mathbb{R}^{n^k \times d}$. Similarly, we say that $f$ is $\mathbb{S}_n$-invariant (or permutation invariant) if and only if $f(\sigma \cdot \mathcal{X}) = f(\mathcal{X})$.*

**Definition 6.** *An $\mathbb{S}_n$-equivariant network is a function $f : \mathbb{R}^{n^k \times d} \to \mathbb{R}^{n^{k'} \times d'}$ defined as a composition of $\mathbb{S}_n$-equivariant linear functions $f_1, .., f_T$ and $\mathbb{S}_n$-equivariant nonlinearities $\gamma_1, .., \gamma_T$:*

$$f = \gamma_T \circ f_T \circ .. \circ \gamma_1 \circ f_1.$$

*On the another hand, an $\mathbb{S}_n$-invariant network is a function $f : \mathbb{R}^{n^k \times d} \to \mathbb{R}$ defined as a composition of an $\mathbb{S}_n$-equivariant network $f'$ and an $\mathbb{S}_n$-invariant function on top of it, e.g., $f = f'' \circ f'$.*

In order to build higher order equivariant networks, we revisit some basic tensor operations: the tensor product $A \otimes B$ and tensor contraction $A_{\downarrow x_1,..,x_p}$ (details and definitions are Sec. A). It can be shown that these tensor operations respect permutation equivariance (Kondor et al., 2018). Based on these tensor contractions and Def. 5, we can construct the second-order $\mathbb{S}_n$-equivariant networks as in Def. 6 (see Example 1): $f = \gamma \circ \mathcal{M}_T \circ .. \circ \gamma \circ \mathcal{M}_1$. The second-order networks are particularly essential for us to extend the original variational autoencoder (VAE) (Kingma & Welling,

2014) model that approximates the posterior distribution by an *isotropic* Gaussian distribution with a diagonal covariance matrix and uses a fixed prior distribution $\mathcal{N}(0,1)$. In constrast, we generalize by modeling the posterior by $\mathcal{N}(\mu, \Sigma)$ in which $\Sigma$ is a full covariance matrix, and we learn an adaptive parameterized prior $\mathcal{N}(\hat{\mu}, \hat{\Sigma})$ instead of a fixed one. Only the second-order encoders can output a permutation equivariant full covariance matrix, while lower-order networks such as MPNNs are unable to. See Sec. 4.2, B and C for details.

**Example 1.** *The second order message passing has the message $\mathcal{H}_0 \in \mathbb{R}^{|\mathcal{V}| \times |\mathcal{V}| \times (d_v + d_e)}$ initialized by promoting the node features $\mathcal{F}_v$ to a second order tensor (e.g., we treat node features as self-loop edge features), and concatenating with the edge features $\mathcal{F}_e$. Iteratively,*

$$\mathcal{H}_t = \gamma(\mathcal{M}_t), \quad \mathcal{M}_t = \mathcal{W}_t \left[ \bigoplus_{i,j} (\mathcal{A} \otimes \mathcal{H}_{t-1})_{\downarrow i,j} \right],$$

*where $\mathcal{A} \otimes \mathcal{H}_{t-1}$ results in a fourth order tensor while $\downarrow_{i,j}$ contracts it down to a second order tensor along the $i$-th and $j$-th dimensions, $\oplus$ denotes concatenation along the feature channels, and $\mathcal{W}_t$ denotes a multilayer perceptron on the feature channels. We remark that the popular MPNNs (Gilmer et al., 2017b) is a lower-order one and a special case in which $\mathcal{M}_t = \mathcal{D}^{-1} \mathcal{A} \mathcal{H}_{t-1} \mathcal{W}_{t-1}$ where $\mathcal{D}_{ii} = \sum_j \mathcal{A}_{ij}$ is the diagonal matrix of node degrees. The message $\mathcal{H}_T$ of the last iteration is still second order, so we contract it down to the first order latent $\mathcal{Z} = \bigoplus_i \mathcal{H}_{T \downarrow i}$.*

### 3.3 LEARNING TO CLUSTER

**Definition 7.** *A clustering of $n$ objects into $k$ clusters is a mapping $\pi : \{1, .., n\} \rightarrow \{1, .., k\}$ in which $\pi(i) = j$ if the $i$-th object is assigned to the $j$-th cluster. The inverse mapping $\pi^{-1}(j) = \{i \in [1, n] : \pi(i) = j\}$ gives the set of all objects assigned to the $j$-th cluster. The clustering is represented by an assignment matrix $\Pi \in \{0, 1\}^{n \times k}$ such that $\Pi_{i, \pi(i)} = 1$.*

**Definition 8.** *The action of $\mathbb{S}_n$ on a clustering $\pi$ of $n$ objects into $k$ clusters and its corresponding assignment matrix $\Pi$ are*

$$[\sigma \cdot \pi](i) = \pi(\sigma^{-1}(i)), \qquad [\sigma \cdot \Pi]_{i,j} = \Pi_{\sigma^{-1}(i),j}, \qquad \sigma \in \mathbb{S}_n.$$

**Definition 9.** *Let $\mathcal{N}$ be a neural network that takes as input a graph $\mathcal{G}$ of $n$ nodes, and outputs a clustering $\pi$ of $k$ clusters. $\mathcal{N}$ is said to be equivariant if and only if $\mathcal{N}(\sigma \cdot \mathcal{G}) = \sigma \cdot \mathcal{N}(\mathcal{G})$ for all $\sigma \in \mathbb{S}_n$.*

From Def. 9, intuitively the assignement matrix $\Pi$ still represents the same clustering if we permute its rows. The learnable clustering procedure $c(\mathcal{G}^{(\ell)}; \boldsymbol{\theta})$ is built as follows:

1. A graph neural network parameterized by $\boldsymbol{\theta}$ encodes graph $\mathcal{G}^{(\ell)}$ into a first order tensor of $K$ feature channels $\tilde{p}^{(\ell)} \in \mathbb{R}^{|\mathcal{V}^{(\ell)}| \times K}$.
2. The clustering assignment is determined by a row-wise maximum pooling operation:

$$\pi^{(\ell)}(i) = \arg\max_{k \in [1,K]} \tilde{p}_{i,k}^{(\ell)} \tag{1}$$

   that is an equivariant clustering in the sense of Def. 9.

A composition of an equivariant function (e.g., graph net) and an equivariant function (e.g., maximum pooling given in Eq. 1) is still an equivariant function with respect to the node permutation. Thus, the learnable clustering procedure $c(\mathcal{G}^{(\ell)}; \boldsymbol{\theta})$ is permutation equivariant.

In practice, in order to make the clustering procedure differentiable for backpropagation, we replace the maximum pooling in Eq. 1 by sampling from a categorical distribution. Let $\pi^{(\ell)}(i)$ be a categorical variable with class probabilities $p_{i,1}^{(\ell)}, .., p_{i,K}^{(\ell)}$ computed as softmax from $\tilde{p}_{i,:}^{(\ell)}$. The Gumbel-max trick (Gumbel, 1954)(Maddison et al., 2014)(Jang et al., 2017) provides a simple and efficient way to draw samples $\pi^{(\ell)}(i)$:

$$\Pi_i^{(\ell)} = \text{one-hot}\left( \arg\max_{k \in [1,K]} \left[ g_{i,k} + \log p_{i,k}^{(\ell)} \right] \right),$$

where $g_{i,1}, .., g_{i,K}$ are i.i.d samples drawn from Gumbel$(0,1)$. Given the clustering assignment matrix $\Pi^{(\ell)}$, the coarsened adjacency matrix $\mathcal{A}^{(\ell-1)}$ (see Defs. 1 and 2) can be constructed as $\Pi^{(\ell)^T} \mathcal{A}^{(\ell)} \Pi^{(\ell)}$.

It is desirable to have a *balanced* $K$-cluster partition in which clusters $\mathcal{V}_1^{(\ell)}, .., \mathcal{V}_K^{(\ell)}$ have similar sizes that are close to $|\mathcal{V}^{(\ell)}|/K$. The local encoders tend to generalize better for same-size subgraphs. We want the distribution of nodes into clusters to be close to the uniform distribution. We enforce the clustering procedure to produce a balanced cut by minimizing the following Kullback–Leibler divergence:

$$\mathcal{D}_{KL}(P\|Q) = \sum_{k=1}^{K} P(k) \log \frac{P(k)}{Q(k)} \quad \text{where} \quad P = \left(\frac{|\mathcal{V}_1^{(\ell)}|}{|\mathcal{V}^{(\ell)}|}, .., \frac{|\mathcal{V}_K^{(\ell)}|}{|\mathcal{V}^{(\ell)}|}\right), \quad Q = \left(\frac{1}{K}, .., \frac{1}{K}\right). \quad (2)$$

The whole construction of MGN is *equivariant* with respect to node permutations of $\mathcal{G}$. In the case of molecular property prediction, we want MGN to learn to predict a real value $y \in \mathbb{R}$ for each graph $\mathcal{G}$ while learning to find a balanced cut in each resolution to construct a hierarchical structure of latents and coarsen graphs. The total loss function is

$$\mathcal{L}_{\text{MGN}}(\mathcal{G}, y) = \left\|f\left(\bigoplus_{\ell=1}^{L} R(\mathcal{Z}^{(\ell)})\right) - y\right\|_2^2 + \sum_{\ell=1}^{L} \lambda^{(\ell)} \mathcal{D}_{\text{KL}}(P^{(\ell)}\|Q^{(\ell)}), \quad (3)$$

where $f$ is a multilayer perceptron, $\oplus$ denotes the vector concatenation, $R$ is a readout function that produces a permutation invariant vector of size $d$ given the latent $\mathcal{Z}^{|\mathcal{V}^{(\ell)}| \times d}$ at the $\ell$-th resolution, $\lambda^{(\ell)} \in \mathbb{R}$ is a hyperparamter, and $\mathcal{D}_{\text{KL}}(P^{(\ell)}\|Q^{(\ell)})$ is the balanced-cut loss as defined in Eq. 2.

## 4 Hierarchical generative model

In this section, we introduce our hierarchical generative model for multiresolution graph generation based on variational principles.

### 4.1 Background on graph variational autoencoder

Suppose that we have input data consisting of $m$ graphs (data points) $\mathcal{G} = \{\mathcal{G}_1, .., \mathcal{G}_m\}$. The standard variational autoencoders (VAEs), introduced by Kingma & Welling (2014) have the following generation process, in which each data graph $\mathcal{G}_i$ for $i \in \{1, 2, .., m\}$ is generated independently:

1. Generate the latent variables $\mathcal{Z} = \{\mathcal{Z}_1, .., \mathcal{Z}_m\}$, where each $\mathcal{Z}_i \in \mathbb{R}^{|\mathcal{V}_i| \times d_z}$ is drawn i.i.d. from a prior distribution $p_0$ (e.g., standard Normal distribution $\mathcal{N}(0, 1)$).
2. Generate the data graph $\mathcal{G}_i \sim p_\theta(\mathcal{G}_i|\mathcal{Z}_i)$ from the model conditional distribution $p_\theta$.

We want to optimize $\theta$ to maximize the likelihood $p_\theta(\mathcal{G}) = \int p_\theta(\mathcal{Z})p_\theta(\mathcal{G}|\mathcal{Z})d\mathcal{Z}$. However, this requires computing the posterior distribution $p_\theta(\mathcal{G}|\mathcal{Z}) = \prod_{i=1}^{m} p_\theta(\mathcal{G}_i|\mathcal{Z}_i)$, which is usually intractable. Instead, VAEs apply the variational principle, proposed by Wainwright & Jordan (2005), to approximate the posterior distribution as $q_\phi(\mathcal{Z}|\mathcal{G}) = \prod_{i=1}^{m} q_\phi(\mathcal{Z}_i|\mathcal{G}_i)$ via amortized inference and maximize the *evidence lower bound* (ELBO) that is a lower bound of the likelihood:

$$\mathcal{L}_{\text{ELBO}}(\phi, \theta) = \mathbb{E}_{q_\phi(\mathcal{Z}|\mathcal{G})}\left[\log p_\theta(\mathcal{G}|\mathcal{Z})\right] - \mathcal{D}_{\text{KL}}(q_\phi(\mathcal{Z}|\mathcal{G})\|p_0(\mathcal{Z}))$$

$$= \sum_{i=1}^{m}\left[\mathbb{E}_{q_\phi(\mathcal{Z}_i|\mathcal{G}_i)}\left[\log p_\theta(\mathcal{G}_i|\mathcal{Z}_i)\right] - \mathcal{D}_{\text{KL}}(q_\phi(\mathcal{Z}_i|\mathcal{G}_i)\|p_0(\mathcal{Z}_i))\right]. \quad (4)$$

The probabilistic encoder $q_\phi(\mathcal{Z}|\mathcal{G})$, the approximation to the posterior of the generative model $p_\theta(\mathcal{G}, \mathcal{Z})$, is modeled using equivariant graph neural networks (see Example 1) as follows. Assume the prior over the latent variables to be the centered isotropic multivariate Gaussian $p_\theta(\mathcal{Z}) = \mathcal{N}(\mathcal{Z}; 0, I)$. We let $q_\phi(\mathcal{Z}_i|\mathcal{G}_i)$ be a multivariate Gaussian with a diagonal covariance structure:

$$\log q_\phi(\mathcal{Z}_i|\mathcal{G}_i) = \log \mathcal{N}(\mathcal{Z}_i; \mu_i, \sigma_i^2 I), \quad (5)$$

where $\mu_i, \sigma_i \in \mathbb{R}^{|\mathcal{V}_i| \times d_z}$ are the mean and standard deviation of the approximate posterior output by two equivariant graph encoders. We sample from the posterior $q_\phi$ by using the reparameterization trick: $\mathcal{Z}_i = \mu_i + \sigma_i \odot \epsilon$, where $\epsilon \sim \mathcal{N}(0, I)$ and $\odot$ is the element-wise product.

On the another hand, the probabilistic decoder $p_\theta(\mathcal{G}_i|\mathcal{Z}_i)$ defines a conditional distribution over the entries of the adjacency matrix $\mathcal{A}_i$: $p_\theta(\mathcal{G}_i|\mathcal{Z}_i) = \prod_{(u,v) \in \mathcal{V}_i^2} p_\theta(\mathcal{A}_{iuv} = 1|\mathcal{Z}_{iu}, \mathcal{Z}_{iv})$. For example, Kipf & Welling (2016) suggests a simple dot-product decoder that is trivially equivariant: $p_\theta(\mathcal{A}_{iuv} = 1|\mathcal{Z}_{iu}, \mathcal{Z}_{iv}) = \gamma(\mathcal{Z}_{iu}^T \mathcal{Z}_{iv})$, where $\gamma$ denotes the sigmoid function.

## 4.2 Multiresolution VAEs

Based on the construction of multiresolution graph network (see Sec. 3.1), the latent variables are partitioned into disjoint groups, $\mathcal{Z}_i = \{\mathcal{Z}_i^{(1)}, \mathcal{Z}_i^{(2)}, .., \mathcal{Z}_i^{(L)}\}$ where $\mathcal{Z}_i^{(\ell)} = \{[\mathcal{Z}_i^{(\ell)}]_k \in \mathbb{R}^{|[\mathcal{V}_i^{(\ell)}]_k| \times d_z}\}_k$ is the set of latents at the $\ell$-th resolution level in which the graph $\mathcal{G}_i^{(\ell)}$ is partitioned into a number of clusters $[\mathcal{G}_i^{(\ell)}]_k$.

In the area of normalzing flows (NFs), Wu et al. (2020) has shown that stochasticity (e.g., a chain of stochastic sampling blocks) overcomes expressivity limitations of NFs. In general, our MGVAE is a stochastic version of the deterministic MGN such that stochastic sampling is applied at each resolution level in a bottom-up manner. The prior (Eq. 6) and the approximate posterior (Eq. 7) are represented by

$$p(\mathcal{Z}_i) = \prod_{\ell=1}^{L} p(\mathcal{Z}_i^{(\ell)}) = \prod_{\ell=1}^{L} \prod_k p([\mathcal{Z}_i^{(\ell)}]_k), \tag{6}$$

$$q_\phi(\mathcal{Z}_i|\mathcal{G}_i) = q_\phi(\mathcal{Z}_i^{(L)}|\mathcal{G}_i^{(L)}) \prod_{\ell=L-1}^{1} q_\phi(\mathcal{Z}_i^{(\ell)}|\mathcal{Z}_i^{(\ell+1)}, \mathcal{G}_i^{(\ell)}), \tag{7}$$

in which each conditional in the approximate posterior are in the form of factorial Normal distributions, in particular

$$q_\phi(\mathcal{Z}_i^{(\ell)}|\mathcal{Z}_i^{(\ell+1)}, \mathcal{G}_i^{(\ell)}) = \prod_k q_\phi([\mathcal{Z}_i^{(\ell)}]_k|\mathcal{Z}_i^{(\ell+1)}, [\mathcal{G}_i^{(\ell)}]_k),$$

where each probabilistic encoder $q_\phi([\mathcal{Z}_i^{(\ell)}]_k|\mathcal{Z}_i^{(\ell+1)}, [\mathcal{G}_i^{(\ell)}]_k)$ operates on a subgraph $[\mathcal{G}_i^{(\ell)}]_k$ as follows:

- The pooling network $\boldsymbol{p}^{(\ell+1)}$ shrinks the latent $\mathcal{Z}_i^{(\ell+1)}$ into the node features of $\mathcal{G}_i^{(\ell)}$ as in the construction of MGN (see Def. 4).
- The local (deterministic) graph encoder $\boldsymbol{d}_{\text{local}}^{(\ell)}$ encodes each subgraph $[\mathcal{G}_i^{(\ell)}]_k$ into a mean vector and a diagonal covariance matrix (see Eq. 5). A second order encoder can produce a positive semidefinite non-diagonal covariance matrix, that can be interpreted as a Gaussian Markov Random Fields (details in Sec. B). The new subgraph latent $[\mathcal{Z}_i^{(\ell)}]_k$ is sampled by the reparameterization trick.

The prior can be either the isotropic Gaussian $\mathcal{N}(0, 1)$ as in standard VAEs, or be implemented as a parameterized Gaussian $\mathcal{N}(\hat{\mu}, \hat{\Sigma})$ where $\hat{\mu}$ and $\hat{\Sigma}$ are learnable equivariant functions (details in Sec. C). The reparameterization trick for conventional $\mathcal{N}(0, 1)$ prior is the same as in Sec. 4.1, while the new one for the generalized and learnable prior $\mathcal{N}(\hat{\mu}, \hat{\Sigma})$ is given in Sec. B. On the another hand, the probabilistic decoder $p_\theta(\mathcal{G}_i^{(1)}, .., \mathcal{G}_i^{(L)}|\mathcal{Z}_i^{(1)}, .., \mathcal{Z}_i^{(L)})$ defines a conditional distribution over all subgraph adjacencies at each resolution level:

$$p_\theta(\mathcal{G}_i^{(1)}, .., \mathcal{G}_i^{(L)}|\mathcal{Z}_i^{(1)}, .., \mathcal{Z}_i^{(L)}) = \prod_\ell p_\theta(\mathcal{G}_i^{(\ell)}|\mathcal{Z}_i^{(\ell)}) = \prod_\ell \prod_k p_\theta([\mathcal{A}_i^{(\ell)}]_k|[\mathcal{Z}_i^{(\ell)}]_k).$$

Extending from Eq. 4, we write our multiresolution variational lower bound $\mathcal{L}_{\text{MGVAE}}(\phi, \theta)$ on $\log p(\mathcal{G})$ compactly as

$$\mathcal{L}_{\text{MGVAE}}(\phi, \theta) = \sum_i \sum_\ell \left[ \mathbb{E}_{q_\phi(\mathcal{Z}_i^{(\ell)}|\mathcal{G}_i^{(\ell)})}[\log p_\theta(\mathcal{G}_i^{(\ell)}|\mathcal{Z}_i^{(\ell)})] - \mathcal{D}_{\text{KL}}(q_\phi(\mathcal{Z}_i^{(\ell)}|\mathcal{G}_i^{(\ell)})\|p_0(\mathcal{Z}_i^{(\ell)})) \right], \tag{8}$$

where the first term denotes the reconstruction loss (e.g., $\|\mathcal{A}_i^{(\ell)} - \hat{\mathcal{A}}_i^{(\ell)}\|$ where $\mathcal{A}_i^{(L)}$ is $\mathcal{G}_i$ itself, $\mathcal{A}_i^{(\ell<L)}$ is the adjacency produced by MGN at level $\ell$, and $\hat{\mathcal{A}}_i^{(\ell)}$ are the reconstructed ones by the decoders); and the second term is indeed $\mathcal{D}_{\text{KL}}\big(\mathcal{N}(\mu_i^{(\ell)}, \Sigma_i^{(\ell)})\|\mathcal{N}(\hat{\boldsymbol{\mu}}^{(\ell)}, \hat{\boldsymbol{\Sigma}}^{(\ell)})\big)$ where $\mu_i^{(\ell)} \in \mathbb{R}^{|\mathcal{V}_i^{(\ell)}| \times d}$ and $\Sigma_i^{(\ell)} \in \mathbb{R}^{|\mathcal{V}_i^{(\ell)}| \times |\mathcal{V}_i^{(\ell)}| \times d}$ are the mean and covariance tensors produced by the $\ell$-th encoder for graph $\mathcal{G}_i$, while $\hat{\boldsymbol{\mu}}^{(\ell)}$ and $\hat{\boldsymbol{\Sigma}}^{(\ell)}$ are learnable ones in an equivariant manner as in Sec. C. In general, the overall optimization is given as follows:

$$\min_{\phi, \theta, \{\hat{\boldsymbol{\mu}}^{(\ell)}, \hat{\boldsymbol{\Sigma}}^{(\ell)}\}_\ell} \mathcal{L}_{\text{MGVAE}}(\phi, \theta; \{\hat{\boldsymbol{\mu}}^{(\ell)}, \hat{\boldsymbol{\Sigma}}^{(\ell)}\}_\ell) + \sum_{i,\ell} \lambda^{(\ell)} \mathcal{D}_{\text{KL}}(P_i^{(\ell)}\|Q_i^{(\ell)}), \tag{9}$$

where $\phi$ denotes all learnable parameters of the encoders, $\theta$ denotes all learnable parameters of the decoders, and $\mathcal{D}_{\text{KL}}(P_i^{(\ell)}\|Q_i^{(\ell)})$ is the balanced-cut loss for graph $\mathcal{G}_i$ at level $\ell$ as defined in Sec. 3.3.

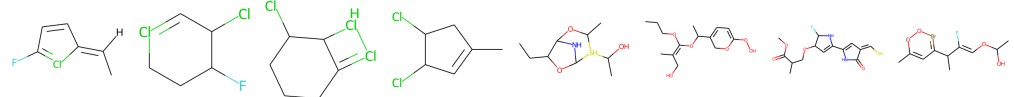

Figure 3: MGVAE generates molecules on QM9 (4 on the left) and ZINC (the rest) equivariantly. There are many more examples of generated molecules in Sec. D.2. Both equivariant MGVAE and autoregressive MGN generate high-quality molecules with complicated structures such as rings.

| Dataset | Method | Training size | Input features | Validity | Novelty | Uniqueness |
|---|---|---|---|---|---|---|
| QM9 | GraphVAE | ~ 100K | Graph | 61.00% | 85.00% | 40.90% |
| | CGVAE | | | 100% | 94.35% | 98.57% |
| | MolGAN | | | 98.1% | 94.2% | 10.4% |
| | **Autoregressive MGN** | 10K | | 100% | 95.01% | 97.44% |
| | **All-at-once MGVAE** | | | 100% | 100% | 95.16% |
| ZINC | GraphVAE | ~ 200K | Graph | 14.00% | 100% | 31.60% |
| | CGVAE | | | 100% | 100% | 99.82% |
| | JT-VAE | | | 100% | - | - |
| | **Autoregressive MGN** | 1K | | 100% | 99.89% | 99.69% |
| | **All-at-once MGVAE** | 10K | Chemical | 99.92% | 100% | 99.34% |

Table 1: Molecular graph generation results. GraphVAE results are taken from (Liu et al., 2018).

## 5 EXPERIMENTS

Many more experimental results and details are presented in the Sec. D of the Appendix.

### 5.1 MOLECULAR GRAPH GENERATION

We examine the generative power of MGN and MGVAE in the challenging task of molecule generation, in which the graphs are highly structured. We demonstrate that MGVAE is the first hierarchical graph VAE model generating graphs in a permutation-equivariant manner that is competitive against autoregressive results. We train on two datasets that are standard in the field:

1. **QM9** (Ruddigkeit et al., 2012) (Ramakrishnan et al., 2014): contains 134K organic molecules with up to nine atoms (C, H, O, N, and F) out of the GDB-17 universe of molecules.
2. **ZINC** (Sterling & Irwin, 2015): contains 250K purchasable drug-like chemical compounds with up to twenty-three heavy atoms.

We only use the graph features as the input, including the adjacency matrix, the one-hot vector of atom types (e.g., carbon, hydrogen, etc.) and the bond types (single bond, double bond, etc.) without any further domain knowledge from chemistry or physics. First, we train autoencoding task of reconstructing the adjacency matrix and node features. We use a learnable equivariant prior (see Sec. C) instead of the conventional $\mathcal{N}(0, 1)$. Then, we generate $5,000$ different samples from the prior, and decode each sample into a generated graph (see Fig. 3). We implement our graph construction (decoding) in two approaches:

1. **All-at-once**: We reconstruct the whole adjacency matrix by running the probabilistic decoder (see Sec. 4). MGVAE enables us to generate a graph at any given resolution level $\ell$. In this particular case, we select the highest resolution $\ell = L$. This approach of decoding preserves equivariance, but is harder to train. On ZINC, we extract several chemical/atomic features from RDKit as the input for the encoders to reach a good convergence in training.
2. **Autoregressive**: The graph is constructed iteratively by adding one edge in each iteration, similarly to (Liu et al., 2018). But this approach does not respect permutation equivariance.

In our setting for small molecules, $L = 3$ and $K = 2^{\ell-1}$ for the $\ell$-th level. We compare our methods with other graph-based generative models including GraphVAE (Simonovsky & Komodakis, 2018), CGVAE (Liu et al., 2018), MolGAN (Cao & Kipf, 2018), and JT-VAE (Jin et al., 2018). We evaluate the quality of generated molecules in three metrics: (i) validity, (ii) novelty and (iii) uniqueness as

| | COMMUNITY-SMALL | | | EGO-SMALL | | |
|---|---|---|---|---|---|---|
| MODEL | DEGREE | CLUSTER | ORBIT | DEGREE | CLUSTER | ORBIT |
| GRAPHVAE | 0.35 | 0.98 | 0.54 | 0.13 | 0.17 | 0.05 |
| DEEPGMG | 0.22 | 0.95 | 0.4 | 0.04 | 0.10 | 0.02 |
| GRAPHRNN | 0.08 | 0.12 | 0.04 | 0.09 | 0.22 | 0.003 |
| GNF | 0.20 | 0.20 | 0.11 | 0.03 | 0.10 | 0.001 |
| GRAPHAF | 0.06 | 0.10 | 0.015 | 0.04 | 0.04 | 0.008 |
| **MGVAE** | **0.002** | **0.01** | **0.01** | **1.74e-05** | **0.0006** | **6.53e-05** |

Table 2: Graph generation results depicting MMD for various graph statistics between the test set and generated graphs. MGVAE outperforms all competing methods.

the percentage of the generated molecules that are chemically valid, different from all molecules in the training set, and not redundant, respectively. Because of high complexity, we only train on a small random subset of examples while all other methods are trained on the full datasets. Our models are equivalent with the state-of-the-art, even with a limited training set (see Table 1). Admittedly, molecule generation is a somewhat subject task that can only be evaluated with objective numerical measures up to a certain point. Qualitatively, however the molecules that MGVAE generates are as good as the state of the art, in some cases better in terms of producing several high-quality drug-like molecules with complicated functional groups and structures. Many further samples generated by MGVAE and their analysis can be found in the Appendix.

## 5.2 GENERAL GRAPH GENERATION BY MGVAE

We further examine the expressive power of hierarchical latent structure of MGVAE in the task of general graph generation. We choose two datasets from GraphRNN paper (You et al., 2018a):

1. **Community-small**: A synthetic dataset of 100 2-community graphs where $12 \le |V| \le 20$.
2. **Ego-small**: 200 3-hop ego networks extracted from the Citeseer network (Sen et al., 2008) where $4 \le |V| \le 18$.

The datasets are generated by the scripts from the GraphRNN codebase (You et al., 2018b). We keep $80\%$ of the data for training and the rest for testing. We evaluate our generated graphs by computing Maximum Mean Discrepancy (MMD) distance between the distributions of graph statistics on the test set and the generated set as proposed by (You et al., 2018a). The graph statistics are node degrees, clustering coefficients, and orbit counts. As suggested by (Liu et al., 2019), we execute 15 runs with different random seeds, and we generate 1,024 graphs for each run, then we average the results over 15 runs. We compare MGVAE against GraphVAE (Simonovsky & Komodakis, 2018), DeepGMG (Li et al., 2018), GraphRNN (You et al., 2018a), GNF (Liu et al., 2019), and GraphAF (Shi et al., 2020). The baselines are taken from GNF paper (Liu et al., 2019) and GraphAF paper (Shi et al., 2020). In our setting of (all-at-once) MGVAE, we implement only $L = 2$ levels of resolution and $K = 2^{\ell}$ clusters for each level. Our encoders have 10 layers of message passing. Instead of using a high order equivariant network as the global decoder for the bottom resolution, we only implement a simple fully connected network that maps the latent $\mathcal{Z}^{(L)} \in \mathbb{R}^{|\mathcal{V}| \times d_z}$ into an adjacency matrix of size $|\mathcal{V}| \times |\mathcal{V}|$. For the ego dataset in particular, we implement the learnable equivariant prior as in Sec. B and Sec.C. Table 2 includes our quantitative results in comparison with other methods. MGVAE outperforms all competing methods. Figs. 10 11 show some generated examples and training examples on the 2-community and ego datasets.

## 6 CONCLUSION

We introduced MGVAE built upon MGN, the first generative model to learn and generate graphs in a multiresolution and equivariant manner. The key idea of MGVAE is learning to construct a series of coarsened graphs along with a hierarchy of latent distributions in the encoding process while learning to decode each latent into the corresponding coarsened graph at every resolution level. MGVAE achieves state-of-the-art results from link prediction to molecule and graph generation, suggesting that accounting for the multiscale structure of graphs is a promising way to make graph neural networks even more powerful.

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

## A    BASIC TENSOR OPERATIONS

In order to build higher order equivariant networks, we revisit some basic tensor operations: tensor product (see Def. 10) and tensor contraction (see Def. 11). It can be shown that these tensor operations respect permutation equivariance. Based on them, we build our second order message passing networks.

**Definition 10.** *The **tensor product** of $A \in \mathbb{R}^{n^a}$ with $B \in \mathbb{R}^{n^b}$ yields a tensor $C = A \otimes B \in \mathbb{R}^{n^{a+b}}$ where*

$$C_{i_1,i_2,..,i_{a+b}} = A_{i_1,i_2,..,i_a} B_{i_{a+1},i_{a+2},..,i_{a+b}}$$

**Definition 11.** *The **contraction** of $A \in \mathbb{R}^{n^a}$ along the pair of dimensions $\{x,y\}$ (assuming $x < y$) yields a $(a-2)$-th order tensor*

$$C_{i_1,..,i_{x-1},j,i_{x+1},..,i_{y-1},j,i_{y+1},..,i_a} = \sum_{i_x,i_y} A_{i_1,..,i_a}$$

*where we assume that $i_x$ and $i_y$ have been removed from amongst the indices of $C$. Using Einstein notation, this can be written more compactly as*

$$C_{\{i_1,i_2,..,i_a\} \setminus \{i_x,i_y\}} = A_{i_1,i_2,..,i_a} \delta^{i_x,i_y}$$

*where $\delta$ is the Kronecker delta. In general, the contraction of $A$ along dimensions $\{x_1,..,x_p\}$ yields a tensor $C = A_{\downarrow x_1,..,x_p} \in \mathbb{R}^{n^{a-p}}$ where*

$$A_{\downarrow x_1,..,x_p} = \sum_{i_{x_1}} \sum_{i_{x_2}} \cdots \sum_{i_{x_p}} A_{i_1,i_2,..,i_a}$$

*or compactly as*

$$A_{\downarrow x_1,..,x_p} = A_{i_1,i_2,..,i_a} \delta^{i_{x_1},i_{x_2},\cdots,i_{x_p}}.$$

## B    MARKOV RANDOM FIELDS

Undirected graphical models have been widely applied in the domains spatial or relational data, such as image analysis and spatial statistics. In general, $k$-th order graph encoders encode an undirected graph $\mathcal{G} = (\mathcal{V}, \mathcal{E})$ into a $k$-th order latent $\boldsymbol{z} \in \mathbb{R}^{n^k \times d_z}$, with learnable parameters $\boldsymbol{\theta}$, can be represented as a parameterized Markov Random Field (MRF) or Markov network. Based on the Hammersley-Clifford theorem (Murphy, 2012) (Koller & Friedman, 2009), a positive distribution $p(\boldsymbol{z}) > 0$ satisfies the conditional independent properties of an undirected graph $\mathcal{G}$ iff $p$ can be represented as a product of potential functions $\psi$, one per *maximal clique*, i.e.,

$$p(\boldsymbol{z}|\boldsymbol{\theta}) = \frac{1}{Z(\boldsymbol{\theta})} \prod_{c \in \mathcal{C}} \psi_c(z_c|\theta_c) \tag{10}$$

where $\mathcal{C}$ is the set of all the (maximal) cliques of $\mathcal{G}$, and $Z(\boldsymbol{\theta})$ is the *partition function* to ensure the overall distribution sums to 1, and given by

$$Z(\boldsymbol{\theta}) = \sum_{\boldsymbol{z}} \prod_{c \in \mathcal{C}} \psi_c(z_c|\theta_c)$$

Eq. 10 can be further written down as

$$p(\boldsymbol{z}|\boldsymbol{\theta}) \propto \prod_{v \in \mathcal{V}} \psi_v(z_v|\boldsymbol{\theta}) \prod_{(s,t) \in \mathcal{E}} \psi_{st}(z_{st}|\boldsymbol{\theta}) \cdots \prod_{c=(i_1,..,i_k) \in \mathcal{C}_k} \psi_c(z_c|\boldsymbol{\theta})$$

where $\psi_v$, $\psi_{st}$, and $\psi_c$ are the first order, second order and $k$-th order outputs of the encoder, corresponding to every vertex in $\mathcal{V}$, every edge in $\mathcal{E}$ and every clique of size $k$ in $\mathcal{C}_k$, respectively. However, factorizing a graph into set of maximal cliques has an exponential time complexity, since the problem of determining if there is a clique of size $k$ in a graph is known as an NP-complete problem. Thus, the factorization based on Hammersley-Clifford theorem is *intractable*. The second order encoder relaxes the restriction of maximal clique into *edges*, that is called as *pairwise* MRF:

$$p(\boldsymbol{z}|\boldsymbol{\theta}) \propto \prod_{s \sim t} \psi_{st}(z_s, z_t)$$

Our second order encoder inherits Gaussian MRF introduced by (Rue & Held, 2005) as *pairwise* MRF of the following form

$$p(\boldsymbol{z}|\boldsymbol{\theta}) \propto \prod_{s \sim t} \psi_{st}(z_s, z_t) \prod_t \psi_t(z_t)$$

where $\psi_{st}(z_s, z_t) = \exp\left(-\frac{1}{2} z_s \Lambda_{st} z_t\right)$ is the edge potential, and $\psi_t(z_t) = \exp\left(-\frac{1}{2}\Lambda_{tt} z_t^2 + \eta_t z_t\right)$ is the vertex potental. The joint distribution can be written in the *information form* of a multivariate Gaussian in which

$$\boldsymbol{\Lambda} = \boldsymbol{\Sigma}^{-1}$$
$$\boldsymbol{\eta} = \boldsymbol{\Lambda}\boldsymbol{\mu}$$
$$p(\boldsymbol{z}|\boldsymbol{\theta}) \propto \exp\left(\boldsymbol{\eta}^T \boldsymbol{z} - \frac{1}{2}\boldsymbol{z}^T \boldsymbol{\Lambda} \boldsymbol{z}\right) \tag{11}$$

Sampling $\boldsymbol{z}$ from $p(\boldsymbol{z}|\boldsymbol{\theta})$ in Eq. 11 is the same as sampling from the multivariate Gaussian $\mathcal{N}(\boldsymbol{\mu}, \boldsymbol{\Sigma})$. To ensure end-to-end equivariance, we set the latent layer to be two tensors $\boldsymbol{\mu} \in \mathbb{R}^{n \times d_z}$ and $\boldsymbol{\Sigma} \in \mathbb{R}^{n \times n \times d_z}$ that corresponds to $d_z$ multivariate Gaussians, whose first index, and second index are first order and second order equivariant with permutations. Computation of $\boldsymbol{\Sigma}$ is trickier than $\boldsymbol{\mu}$, simply because $\boldsymbol{\Sigma}$ must be invertible to be a covariance matrix. Thus, our second order encoder produces tensor $\boldsymbol{L}$ as the second order activation, and set $\boldsymbol{\Sigma} = \boldsymbol{L}\boldsymbol{L}^T$. The reparameterization trick from Kingma & Welling (2014) is changed to

$$\boldsymbol{z} = \boldsymbol{\mu} + \boldsymbol{L}\boldsymbol{\epsilon}, \qquad \boldsymbol{\epsilon} \sim \mathcal{N}(0, 1)$$

## C  EQUIVARIANT LEARNABLE PRIOR

The original VAE published by Kingma & Welling (2014) limits each covariance matrix $\boldsymbol{\Sigma}$ to be diagonal and the prior to be $\mathcal{N}(0, 1)$. Our second order encoder removes the diagonal restriction on the covariance matrix. Furthermore, we allow the prior $\mathcal{N}(\hat{\boldsymbol{\mu}}, \hat{\boldsymbol{\Sigma}})$ to be *learnable* in which $\hat{\boldsymbol{\mu}}$ and $\hat{\boldsymbol{\Sigma}}$ are parameters optimized by back propagation in a data driven manner. Importantly, $\hat{\boldsymbol{\Sigma}}$ cannot be learned directly due to the invertibility restriction. Instead, similarly to the second order encoder, a matrix $\hat{\boldsymbol{L}}$ is optimized, and the prior covariance matrix is constructed by setting $\hat{\boldsymbol{\Sigma}} = \hat{\boldsymbol{L}}\hat{\boldsymbol{L}}^T$. The Kullback-Leibler divergence between the two distributions $\mathcal{N}(\boldsymbol{\mu}, \boldsymbol{\Sigma})$ and $\mathcal{N}(\hat{\boldsymbol{\mu}}, \hat{\boldsymbol{\Sigma}})$ is as follows:

$$\mathcal{D}_{KL}(\mathcal{N}(\boldsymbol{\mu}, \boldsymbol{\Sigma})\|\mathcal{N}(\hat{\boldsymbol{\mu}}, \hat{\boldsymbol{\Sigma}})) = \frac{1}{2}\left(\text{tr}(\hat{\boldsymbol{\Sigma}}^{-1}\boldsymbol{\Sigma}) + (\hat{\boldsymbol{\mu}} - \boldsymbol{\mu})^T \hat{\boldsymbol{\Sigma}}^{-1}(\hat{\boldsymbol{\mu}} - \boldsymbol{\mu}) - n + \ln\left(\frac{\det\hat{\boldsymbol{\Sigma}}}{\det\boldsymbol{\Sigma}}\right)\right) \tag{12}$$

Even though $\boldsymbol{\Sigma}$ is invertible, but gradient computation through the KL-divergence loss can be numerical instable because of Cholesky decomposition procedure in matrix inversion. Thus, we add neglectable noise $\epsilon = 10^{-4}$ to the diagonal of both covariance matrices.

Importantly, during training, the KL-divergence loss breaks the permutation equivariance. Suppose the set of vertices are permuted by a permutation matrix $\boldsymbol{P}_\sigma$ for $\sigma \in \mathbb{S}_n$. Since $\boldsymbol{\mu}$ and $\boldsymbol{\Sigma}$ are the first order and second order equivariant outputs of the encoder, they are changed to $\boldsymbol{P}_\sigma \boldsymbol{\mu}$ and $\boldsymbol{P}_\sigma \boldsymbol{\Sigma} \boldsymbol{P}_\sigma^T$ accordingly. But

$$\mathcal{D}_{\text{KL}}(\mathcal{N}(\boldsymbol{\mu}, \boldsymbol{\Sigma})\|\mathcal{N}(\hat{\boldsymbol{\mu}}, \hat{\boldsymbol{\Sigma}})) \neq \mathcal{D}_{\text{KL}}(\mathcal{N}(\boldsymbol{P}_\sigma\boldsymbol{\mu}, \boldsymbol{P}_\sigma\boldsymbol{\Sigma}\boldsymbol{P}_\sigma^T)\|\mathcal{N}(\hat{\boldsymbol{\mu}}, \hat{\boldsymbol{\Sigma}}))$$

To address the equivariance issue, we want to solve the following convex optimization problem that is our new equivariant loss function

$$\min_{\sigma \in \mathbb{S}_n} \quad \mathcal{D}_{\text{KL}}(\mathcal{N}(\boldsymbol{P}_\sigma\boldsymbol{\mu}, \boldsymbol{P}_\sigma\boldsymbol{\Sigma}\boldsymbol{P}_\sigma^T)\|\mathcal{N}(\hat{\boldsymbol{\mu}}, \hat{\boldsymbol{\Sigma}})) \tag{13}$$

However, solving the optimization based on Eq. 13 is computationally expensive. One solution is to solve the minimum-cost maximum-matching in a bipartite graph (Hungarian matching) with the cost matrix $\boldsymbol{C}_{ij} = \|\mu_i - \hat{\mu}_j\|$ by $O(n^4)$ algorithm published by Edmonds & Karp (1972), that can be still improved further into $O(n^3)$. The Hungarian matching preserves equiariance, but is still computationally expensive. In practice, instead of finding a optimal permutation, we apply a free-matching scheme to find an assignment matrix $\boldsymbol{\Pi}$ such that: $\boldsymbol{\Pi}_{ij^*} = 1$ if and only if $j^* = \arg\min_j \|\mu_i - \hat{\mu}_j\|$, for each $i \in [1, n]$. The free-matching scheme preserves equivariance and can be done efficiently in a simple $O(n^2)$ algorithm that is also suitable for GPU computation.

# D EXPERIMENTS

## D.1 LINK PREDICTION ON CITATION GRAPHS

We demonstrate the ability of the MGVAE models to learn meaningful latent embeddings on a link prediction task on popular citation network datasets Cora and Citeseer (Sen et al., 2008). At training time, 15% of the citation links (edges) were removed while all node features are kept, the models are trained on an incomplete graph Laplacian constructed from the remaining 85% of the edges. From previously removed edges, we sample the same number of pairs of unconnected nodes (non-edges). We form the validation and test sets that contain 5% and 10% of edges with an equal amount of non-edges, respectively.

We compare our model MGVAE against popular methods in the field:

1. Spectral clustering (SC) (Tang & Liu, 2011)
2. Deep walks (DW) (Perozzi et al., 2014)
3. Variational graph autoencoder (VGAE) (Kipf & Welling, 2016)

on the ability to correctly classify edges and non-edges using two metrics: area under the ROC curve (AUC) and average precision (AP). Numerical results of SC and DW are experimental settings are taken from (Kipf & Welling, 2016). We reran the implementation of VGAE as in (Kipf & Welling, 2016).

For MGVAE, we initialize weights by Glorot initialization (Glorot & Bengio, 2010). We repeat the experiments with 5 different random seeds and calculate the average AUC and AP along with their standard deviations. The number of message passing layers ranges from 1 to 4. The size of latent representation is 128. The number of coarsening levels is $L \in \{3, 7\}$. In the $\ell$-th coarsening level, we partition the graph $\mathcal{G}^{(\ell)}$ into $2^\ell$ (for $L = 7$) or $4^\ell$ (for $L = 3$) clusters. We train for 2,048 epochs using Adam optimization (Kingma & Ba, 2015) with a starting learning rate of $0.01$. Hyperparameters optimization (e.g. number of layers, dimension of the latent representation, etc.) is done on the validation set. MGVAE outperforms all other methods (see Table 3).

We propose our learning to cluster algorithm to achieve the balanced $K$-cut at every resolution level. Besides, we also implement two fixed clustering algorithms:

1. **Spectral**: It is similar to the one implemented in (Rustamov & Guibas, 2013).
   - First, we embed each node $i \in \mathcal{V}$ into $\mathbb{R}^{n_{max}}$ as $(\xi_1(i)/\lambda_1(i), .., \xi_{n_{max}}(i)/\lambda_{n_{max}}(i))$, where $\{\lambda_n, \xi_n\}_{n=0}^{n_{max}}$ are the eigen-pairs of the graph Laplacian $\mathcal{L} = \mathcal{D}^{-1}(\mathcal{D} - \mathcal{A})$ where $\mathcal{D}_{ii} = \sum_j \mathcal{A}_{ij}$. We assume that $\lambda_0 \leq .. \leq \lambda_{n_{max}}$. In this case, $n_{max} = 10$.
   - At the $\ell$-th resolution level, we apply the K-Means clustering algorithm based on the above node embedding to partition graph $\mathcal{G}^{(\ell)}$.

2. **K-Means**:
   - First, we apply PCA to compress the sparse word frequency vectors (of size 1,433 on Cora and 3,703 on Citeseer) associating with each node into 10 dimensions.
   - We use the compressed node embedding for the K-Means clustering.

Tables 4 and 5 show that our learning to cluster algorithm returns a much more balanced cut on the highest resolution level comparing to both Spectral and K-Means clusterings. For instance, we

Table 3: Citation graph link prediction results (AUC & AP)

| Dataset | Cora | | Citeseer | |
|---|---|---|---|---|
| **Method** | AUC (ROC) | AP | AUC (ROC) | AP |
| SC | 84.6 ± 0.01 | 88.5 ± 0.00 | 80.5 ± 0.01 | 85.0 ± 0.01 |
| DW | 83.1 ± 0.01 | 85.0 ± 0.00 | 80.5 ± 0.02 | 83.6 ± 0.01 |
| VGAE | 90.97 ± 0.77 | 91.88 ± 0.83 | 89.63 ± 1.04 | 91.10 ± 1.02 |
| **MGVAE** (Spectral) | 91.19 ± 0.76 | 92.27 ± 0.73 | 90.55 ± 1.17 | 91.89 ± 1.27 |
| **MGVAE** (K-Means) | 93.07 ± 5.61 | 92.49 ± 5.77 | 90.81 ± 1.19 | 91.98 ± 1.02 |
| **MGVAE** | **95.67 ± 3.11** | **95.02 ± 3.36** | **93.93 ± 5.87** | **93.06 ± 6.33** |

| Method | Min | Max | STD | KL divergence |
|---|---|---|---|---|
| Spectral | 1 | 2020 | 177.52 | 3.14 |
| K-Means | 1 | 364 | 40.17 | 0.84 |
| Learn to cluster | 10 | 36 | 4.77 | **0.02** |

Table 4: Learning to cluster algorithm returns balanced cuts on Cora.

| Method | Min | Max | STD | KL divergence |
|---|---|---|---|---|
| Spectral | 1 | 3320 | 292.21 | 4.51 |
| K-Means | 1 | 326 | 41.69 | 0.74 |
| Learn to cluster | 11 | 38 | 4.93 | **0.01** |

Table 5: Learning to cluster algorithm returns balanced cuts on Citeseer.

have $L = 7$ resolution levels and we partition the $\ell$-th resolution into $K = 2^\ell$ clusters. Thus, on the bottom levels, we have 128 clusters. If we distribute nodes into clusters uniformly, the expected number of nodes in a cluster is $21.15$ and $25.99$ on Cora ($2,708$ nodes) and Citeseer ($3,327$ nodes), respectively. We measure the minimum, maximum, standard deviation of the numbers of nodes in 128 clusters. Furthermore, we measure the Kullback–Leibler divergence between the distribution of nodes into clusters and the uniform distribution. Our learning to cluster algorithm achieves low KL losses of $0.02$ and $0.01$ on Cora and Citeseer, respectively.

## D.2 Molecular graph generation

In this case, MGVAE and MGN are implemented with $L = 3$ resolution levels, and the $\ell$-th resolution graph is partitioned into $K = 2^{\ell-1}$ clusters. On each resolution level, the local encoders and local decoders are second-order $\mathbb{S}_n$-equivariant networks with up to 4 equivariant layers. The number of channels for each node latent $d_z$ is set to 256. We apply two approaches for graph decoding:

1. **All-at-once**: MGVAE reconstructs all resolution adjacencies by equivariant decoder networks. Furthermore, we apply learnable equivariant prior as in Sec. C. Our second order encoders are interpreted as Markov Random Fields (see Sec. B). This approach preserves permutation equivariance. In addition, we implement a correcting process: the decoder network of the highest resolution level returns a probability for each edge, we sort these probabilities in a descending order and gradually add the edges in that order to satisfy all chemical constraints. Furthermore, we investigate the expressive power of the second order $\mathbb{S}_n$-equivariant decoder by replacing it by a multilayer perceptron (MLP) decoder with 2 hidden layers of size 512 and sigmoid nonlinearity. We find that the higher order decoder outperforms the MLP decoder given the same encoding architecture. Table 6 shows the comparison between the two decoding models.

2. **Autoregressive**: This decoding process is constructed in an autoregressive manner similarly to (Liu et al., 2018). First, we sample each vertex latent $z$ independently. We randomly select a starting node $v_0$, then we apply Breath First Search (BFS) to determine a particular node ordering from the node $v_0$, however that breaks the permutation equivariance. Then iteratively we add/sample new edge to the existing graph $\mathcal{G}_t$ at the $t$-th iteration (given a randomly selected node $v_0$ as the start graph $\mathcal{G}_0$) until completion. We apply second-order MGN with gated recurrent architecture to produce the probability of edge $(u, v)$ where one vertex $u$ is in the existing graph $\mathcal{G}_t$ and the another one is outside; and also the probability of its label. Intuitively, the decoding process is a sequential classification.

We randomly select 10,000 training examples for QM9; and 1,000 (autoregressive) and 10,000 (all-at-once) training examples for ZINC. It is important to note that our training sets are much smaller comparing to other methods. For all of our generation experiments, we only use graph features as the input for the encoder such as one-hot atomic types and bond types. Since ZINC molecules are larger then QM9 ones, it is more difficult to train with the second order $\mathbb{S}_n$-equivariant decoders (e.g., the number of bond/non-bond predictions or the number of entries in the adjacency matrices are proportional to squared number of nodes). Therefore, we input several chemical/atomic features

| Dataset | Method | Validity | Novelty | Uniqueness |
|---------|--------|----------|---------|------------|
| QM9 | MLP decoder | 100% | 99.98% | 77.62% |
| | $\mathbb{S}_n$ decoder | 100% | 100% | 95.16% |

Table 6: All-at-once MGVAE with MLP decoder vs. second order decoder.

| Feature | Type | Number | Description |
|---------|------|--------|-------------|
| `GetAtomicNum` | Integer | 1 | Atomic number |
| `IsInRing` | Boolean | 1 | Belongs to a ring? |
| `IsInRingSize` | Boolean | 9 | Belongs to a ring of size $k \in \{1, .., 9\}$? |
| `GetIsAromatic` | Boolean | 1 | Aromaticity? |
| `GetDegree` | Integer | 1 | Vertex degree |
| `GetExplicitValance` | Integer | 1 | Explicit valance |
| `GetFormalCharge` | Integer | 1 | Formal charge |
| `GetIsotope` | Integer | 1 | Isotope |
| `GetMass` | Double | 1 | Atomic mass |
| `GetNoImplicit` | Boolean | 1 | Allowed to have implicit Hs? |
| `GetNumExplicitHs` | Integer | 1 | Number of explicit Hs |
| `GetNumImplicitHs` | Integer | 1 | Number of implicit Hs |
| `GetNumRadicalElectrons` | Integer | 1 | Number of radical electrons |
| `GetTotalDegree` | Integer | 1 | Total degree |
| `GetTotalNumHs` | Integer | 1 | Total number of Hs |
| `GetTotalValence` | Integer | 1 | Total valance |

Table 7: The list of chemical/atomic features used for the all-at-once MGVAE on ZINC. We denote each feature by its API in RDKit.

computed from RDKit for the all-at-once MGVAE on ZINC (see Table 7). We concatenate all these features into a vector of size 24 for each atom.

We train our models with Adam optimization method (Kingma & Ba, 2015) with the initial learning rate of $10^{-3}$. Figs. 4 and 5 show some selected examples out of 5,000 generated molecules on QM9 by all-at-once MGVAE, while Fig. 6 shows the molecules generated by autoregressive MGN. Qualitatively, both the decoding approaches capture similar molecular substructures (bond structures). Fig. 9 shows an example of interpolation on the latent space on ZINC with the all-at-once MGVAE. Fig. 7 shows some generated molecules on ZINC by the all-at-once MGVAE. Fig. 8 and table 8 show some generated molecules by the autoregressive MGN on ZINC dataset with high Quantitative Estimate of Drug-Likeness (QED) computed by RDKit and their SMILES strings. On ZINC, the average QED score of the generated molecules is $0.45$ with standard deviation $0.21$. On QM9, the QED score is $0.44 \pm 0.07$.

### D.3  GENERAL GRAPH GENERATION BY MGVAE

Figs. 10 11 show some generated examples and training examples on the 2-community and ego datasets, respectively.

### D.4  UNSUPERVISED MOLECULAR PROPERTIES PREDICTION ON QM9

Density Function Theory (DFT) is the most successful and widely used approach of modern quantum chemistry to compute the electronic structure of matter, and to calculate many properties of molecular systems with high accuracy (Hohenberg & Kohn, 1964). However, DFT is computationally expensive (Gilmer et al., 2017a), that leads to the use of machine learning to estimate the properties of compounds from their chemical structure rather than computing them explicitly with DFT (Hy et al., 2018). To demonstrate that MGVAE can learn a useful molecular representations and capture important molecular structures in an unsupervised and variational autoencoding manner, we extract the highest resolution latents (at $\ell = L$) and use them as the molecular representations for the downstream tasks of predicting DFT's molecular properties on QM9 including 13 learning

Figure 4: Some generated examples on QM9 by the all-at-once MGVAE with second order $\mathbb{S}_n$-equivariant decoders.

Figure 5: Some generated examples on QM9 by the all-at-once MGVAE with a MLP decoder instead of the second order $\mathbb{S}_n$-equivariant one. It generates more tree-like structures.

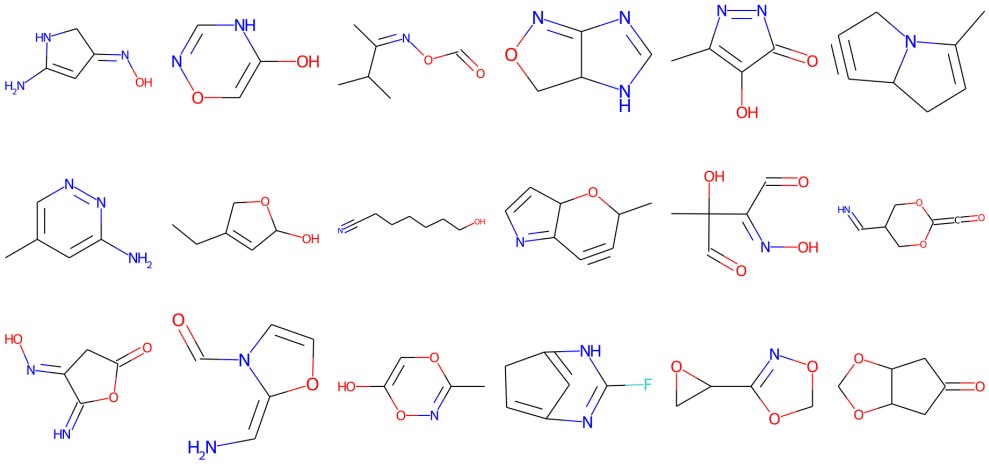

Figure 6: Some generated examples on QM9 by the autoregressive MGN.

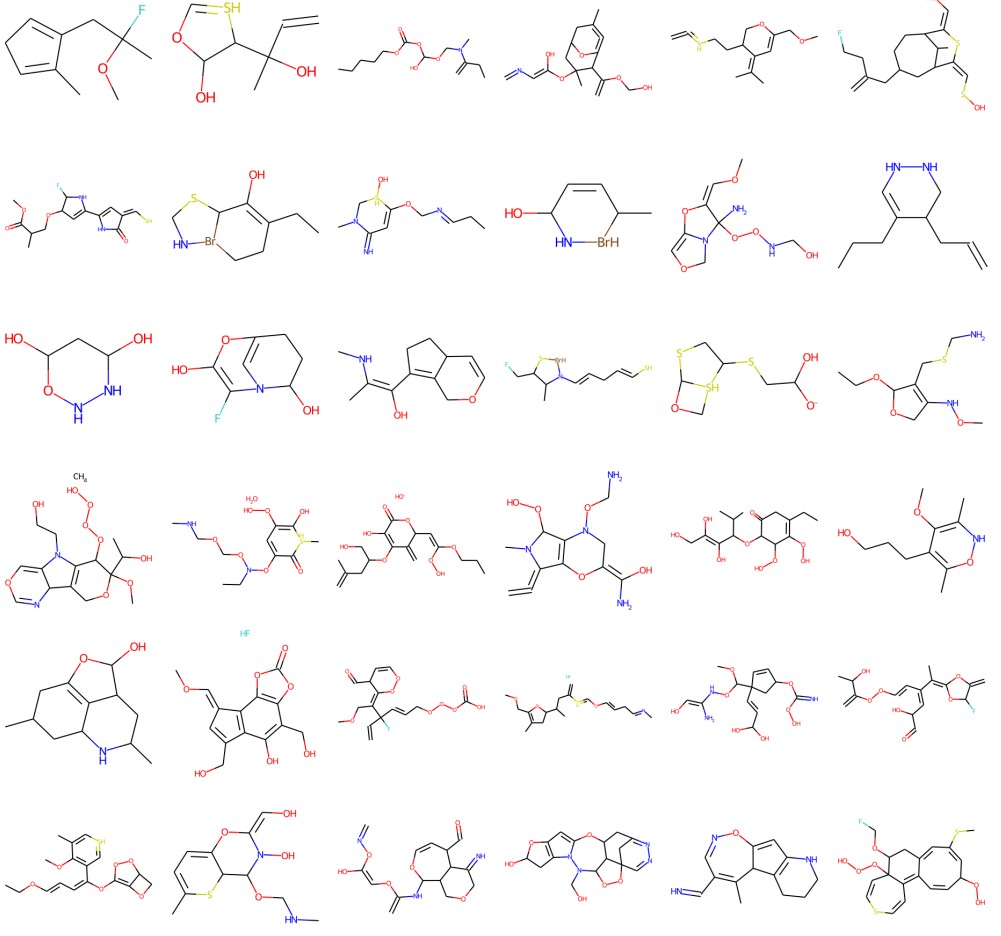

Figure 7: Some generated examples on ZINC by the all-at-once MGVAE with second order $\mathbb{S}_n$-equivariant decoders. In addition of graph features such as one-hot atomic types, we include several chemical features computed from RDKit (as in Table 7) as the input for the encoders. A generated example can contain more than one connected components, each of them is a valid molecule.

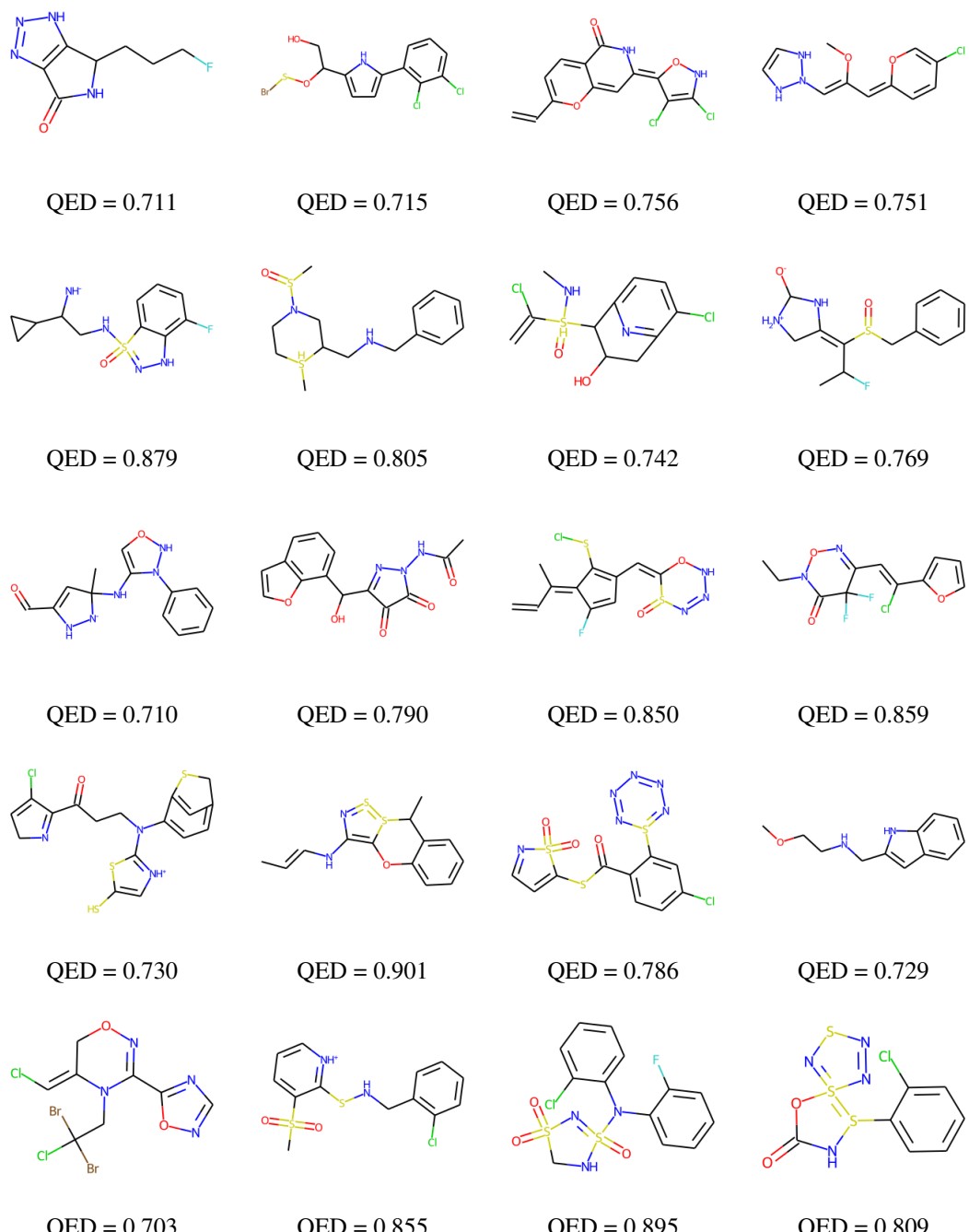

QED = 0.711   QED = 0.715   QED = 0.756   QED = 0.751

QED = 0.879   QED = 0.805   QED = 0.742   QED = 0.769

QED = 0.710   QED = 0.790   QED = 0.850   QED = 0.859

QED = 0.730   QED = 0.901   QED = 0.786   QED = 0.729

QED = 0.703   QED = 0.855   QED = 0.895   QED = 0.809

Figure 8: Some generated molecules on ZINC by the autoregressive MGN with high QED (drug-likeness score).

| Row | Column | SMILES |
|---|---|---|
| 1 | 1 | O=C1NC(CCCF)c2[nH]nnc21 |
| | 2 | OCC(OSBr)c1ccc(-c2cccc(Cl)c2Cl)[nH]1 |
| | 3 | C=CC1=CC=c2c(cc(=C3ONC(Cl)=C3Cl)[nH]c2=O)O1 |
| | 4 | COC(=CN1NC=CN1)C=C1C=CC(Cl)=CO1 |
| 2 | 1 | [NH-]C(CNS1(=O)=NNc2c(F)cccc21)C1CC1 |
| | 2 | CS(=O)N1CC[SH](C)C(CNCc2ccccc2)C1 |
| | 3 | C=C(Cl)[SH](=O)(NC)C1c2ccc(Cl)c(n2)CC1O |
| | 4 | CC(F)C(=C1C[NH2+]C([O-])N1)S(=O)Cc1ccccc1 |
| 3 | 1 | CC1(NC2=CONN2c2ccccc2)C=C(C=O)N[N-]1 |
| | 2 | CC(=O)NN1N=C(C(O)c2cccc3ccoc23)C(=O)C1=O |
| | 3 | C=CC(C)=C1C(F)=CC(C=C2ONN=NS2=O)=C1SCl |
| | 4 | CCN1ON=C(C=C(Cl)c2ccco2)C(F)(F)C1=O |
| 4 | 1 | O=C(CCN(c1[nH+]cc(S)s1)c1ccc2cc1SC2)C1=NCC=C1Cl |
| | 2 | CC=CNC1=C2Oc3ccccc3C(C)S2=S=N1 |
| | 3 | O=C(SC1=CC=NS1(=O)=O)c1ccc(Cl)cc1S1=NN=NN=N1 |
| | 4 | COCCNCc1cc2ccccc2[nH]1 |
| 5 | 1 | ClC=C1CON=C(c2ncno2)N1CC(Cl)(Br)Br |
| | 2 | CS(=O)(=O)c1ccc[nH+]c1SNCc1ccccc1Cl |
| | 3 | O=S1(=O)CNS(=O)(N(c2ccccc2F)c2ccccc2Cl)=N1 |
| | 4 | O=C1NS(c2ccccc2Cl)=S2(=NSN=N2)O1 |

Table 8: SMILES of the generated molecules included in Fig. 8. Online drawing tool: `https://pubchem.ncbi.nlm.nih.gov//edit3/index.html`

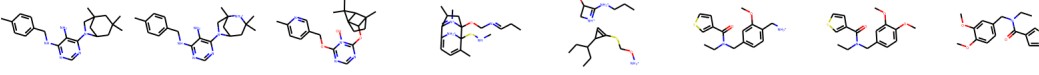

Figure 9: Interpolation on the latent space: we randomly select two molecules from ZINC and we reconstruct the corresponding molecular graphs on the interpolation line between the two latents.

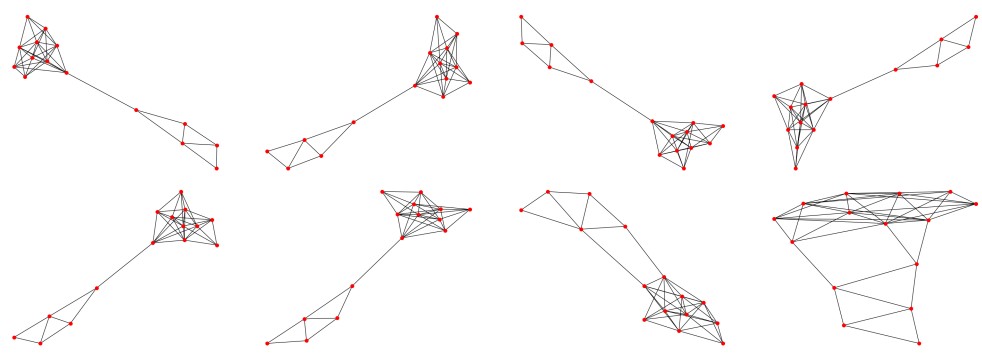

Figure 10: The top row includes generated examples and the bottom row includes training examples on the synthetic 2-community dataset.

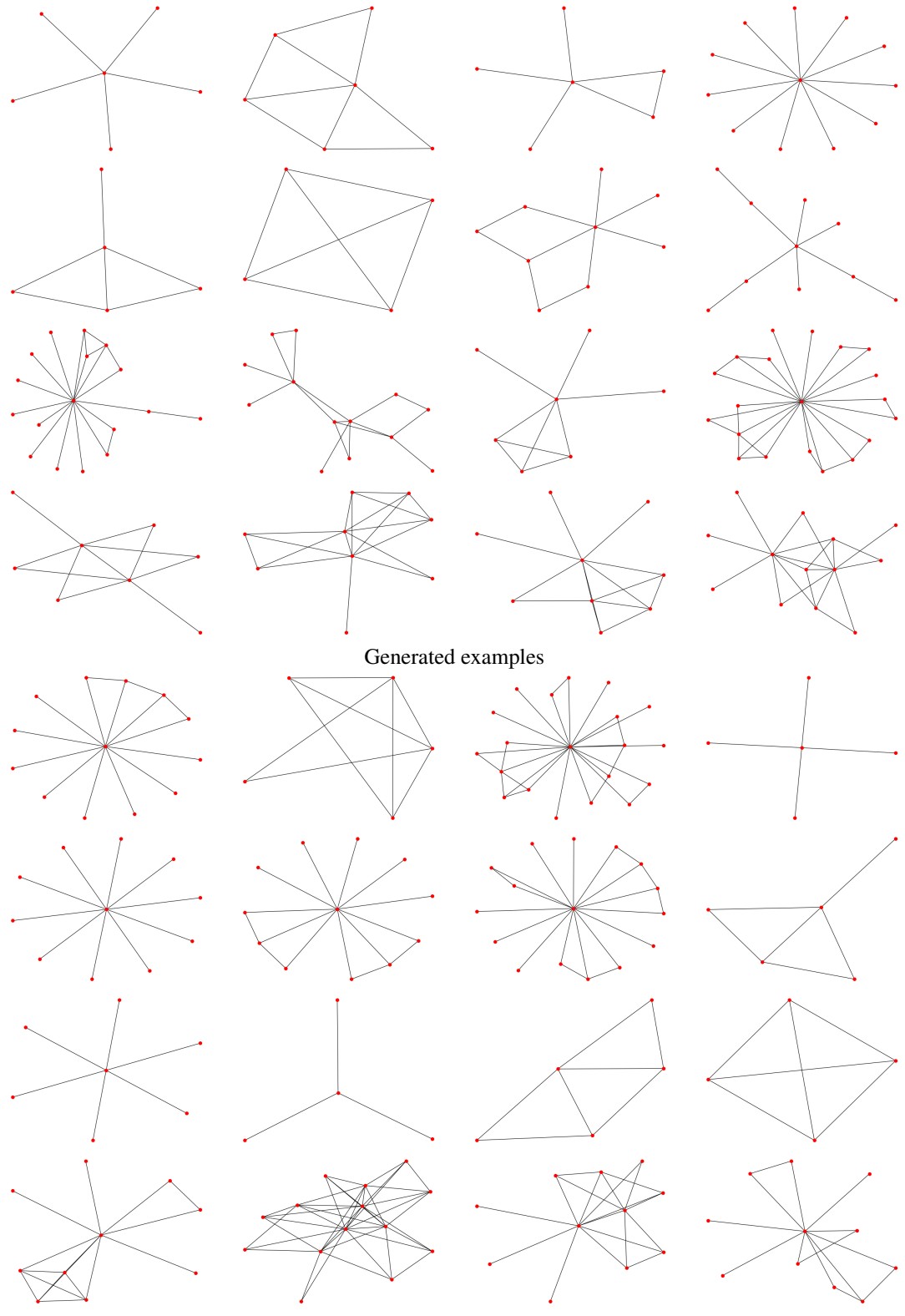

Generated examples

Training examples

Figure 11: **EGO-SMALL**.

| Target | Unit | Mean | STD | Description |
|--------|------|------|-----|-------------|
| $\alpha$ | $bohr^3$ | 75.2808 | 8.1729 | Norm of the static polarizability |
| $C_v$ | cal/mol/K | 31.6204 | 4.0674 | Heat capacity at room temperature |
| G | eV | -70.8352 | 9.4975 | Free energy of atomization |
| gap | eV | 6.8583 | 1.2841 | Difference between HOMO and LUMO |
| H | eV | -77.0167 | 10.4884 | Enthalpy of atomization at room temperature |
| HOMO | eV | -6.5362 | 0.5977 | Highest occupied molecular orbital |
| LUMO | eV | 0.3220 | 1.2748 | Lowest unoccupied molecular orbital |
| $\mu$ | D | 2.6729 | 1.5034 | Norm of the dipole moment |
| $\omega_1$ | $cm^{-1}$ | 3504.1155 | 266.8982 | Highest fundamental vibrational frequency |
| $R^2$ | $bohr^2$ | 1189.4091 | 280.4725 | Electronic spatial extent |
| U | eV | -76.5789 | 10.4143 | Atomization energy at room temperature |
| $U_0$ | eV | -76.1145 | 10.3229 | Atomization energy at 0 K |
| ZPVE | eV | 4.0568 | 0.9016 | Zero point vibrational energy |

Table 9: Description and statistics of 13 learning targets on QM9.

| | alpha | Cv | G | gap | H | HOMO | LUMO | mu | omega1 | R2 | U | U0 | ZPVE |
|--|-------|-----|-----|------|------|-------|-------|------|--------|------|------|------|-------|
| WL | 3.75 | 2.39 | 4.84 | 0.92 | 5.45 | 0.38 | 0.89 | 1.03 | 192 | 154 | 5.41 | 5.36 | 0.51 |
| NGF | 3.51 | 1.91 | 4.36 | 0.86 | 4.92 | 0.34 | 0.82 | 0.94 | 168 | 137 | 4.89 | 4.85 | 0.45 |
| PSCN | 1.63 | 1.09 | 3.13 | 0.77 | 3.56 | 0.30 | 0.75 | 0.81 | 152 | 61 | 3.54 | 3.50 | 0.38 |
| CCN 2D | **1.30** | 0.93 | 2.75 | 0.69 | 3.14 | **0.23** | 0.67 | **0.72** | **120** | **53** | 3.02 | 2.99 | 0.35 |
| **MGVAE** | 2.83 | **0.91** | **1.78** | **0.66** | **1.87** | 0.34 | **0.58** | 0.95 | 195 | 90 | **1.89** | **1.90** | **0.14** |

Table 10: Unsupervised molecular representation learning by MGVAE to predict molecular properties calculated by DFT on QM9 dataset.

targets. For the training, we normalize all learning targets to have mean 0 and standard deviation 1. The name, physical unit, and statistics of these learning targets are detailed in Table 9.

The implementation of MGVAE is the same as detailed in Sec. D.2. MGVAE is trained to reconstruct the highest resolution (input) adjacency, its coarsening adjacencies and the node atomic features. In this case, we do not use any chemical features: the node atomic features are just one-hot atomic types. After MGVAE is converged, to obtain the $\mathbb{S}_n$-invariant molecular representation, we average the node latents at the $L$-th level into a vector of size 256. Finally, we apply a simple Multilayer Perceptron with 2 hidden layers of size 512, sigmoid nonlinearity and a linear layer on top to predict the molecular properties based on the extracted molecular representation. We compare the results in Mean Average Error (MAE) in the corresponding physical units with four methods on the same split of training and testing from (Hy et al., 2018):

1. Support Vector Machine on optimal-assignment Weisfeiler-Lehman (WL) graph kernel (Shervashidze et al., 2011) (Kriege et al., 2016)

2. Neural Graph Fingerprint (NGF) (Duvenaud et al., 2015)

3. PATCHY-SAN (PSCN) (Niepert et al., 2016a)

4. Second order $\mathbb{S}_n$-equivariant Covariant Compositional Networks (CCN 2D) (Kondor et al., 2018) (Hy et al., 2018)

Our unsupervised results show that MGVAE is able to learn a universal molecular representation in an unsupervised manner and outperforms WL in 12, NGF in 10, PSCN in 8, and CCN 2D in 8 out of 13 learning targets, respectively (see Table 10). There are other recent methods in the field that use several chemical and geometric information but comparing to them would be unfair.

D.5 SUPERVISED MOLECULAR PROPERTIES PREDICTION ON ZINC

To further demonstrate the comprehensiveness of MGN, we apply our model in a supervised regression task to predict the solubility (LogP) on the ZINC dataset. We use the same split of 10K/1K/1K

| Method | MLP | GCN | GAT | MoNet | DiscenGCN | FactorGCN | GatedGCN$_E$ | **MGN** |
|--------|-----|-----|-----|-------|-----------|-----------|--------------|---------|
| **MAE** | 0.667 | 0.503 | 0.479 | 0.407 | 0.538 | 0.366 | 0.363 | **0.290** |

Table 11: Supervised MGN to predict solubility on ZINC dataset.

for training/validation/testing as in (Dwivedi et al., 2020). The implementation of MGN is almost the same as detailed in Sec. D.2, except we include the latents of all resolution levels into the prediction. In particular, in each resolution level, we average all the node latents into a vector of size 256; then we concatentate all these vectors into a long vector of size $256 \times L$ and apply a linear layer for the regression task. The baseline results are taken from (Yang et al., 2020) including:

1. Multilayer Perceptron (MLP),
2. Graph Convolution Networks (GCN),
3. Graph Attention Networks (GAT) (Velikovi et al., 2018),
4. MoNet (Monti et al., 2017),
5. Disentangled Graph Convolutional Networks (DisenGCN) (Ma et al., 2019),
6. Factorizable Graph Convolutional Networks (FactorGCN) (Yang et al., 2020),
7. GatedGCN$_E$ (Dwivedi et al., 2020) that uses additional edge information.

Our supervised result shows that MGN outperforms the state-of-the-art models in the field with a margin of $20\%$ (see Table 11).

## D.6  GRAPH-BASED IMAGE GENERATION BY MGVAE

In this additional experiment, we apply MGVAE into the task of image generation. Instead of matrix representation, an image $I \in \mathbb{R}^{H \times W}$ is represented by a grid graph of $H \cdot W$ nodes in which each node represents a pixel, each edge is between two neighboring pixels, and each node feature is the corresponding pixel's color (e.g., $\mathbb{R}^1$ in gray scale, and $\mathbb{R}^3$ in RGB scale). Fig. 12 demonstrates an exmaple of graph representation for images. Since images have natural spatial clustering, instead of learning to cluster, we implement a fixed clustering procedure as follows:

- For the $\ell$-th resolution level, we divide the grid graph of size $H^{(\ell)} \times W^{(\ell)}$ into clusters of size $h \times w$ that results into a grid graph of size $\frac{H^{(\ell)}}{h} \times \frac{W^{(\ell)}}{w}$, supposingly $h$ and $w$ are divisible by $H^{(\ell)}$ and $W^{(\ell)}$, respectively. Each resolution is associated with an image $I^{(\ell)}$ that is a zoomed out version of $I^{(\ell+1)}$.

- The global encoder $e^{(\ell)}$ is implemented with 10 layers of message passing that operates on the whole $H^{(\ell)} \times W^{(\ell)}$ grid graph. We sum up all the node latents into a single latent vector $Z^{(\ell)} \in \mathbb{R}^{d_z}$. The global decoder $d^{(\ell)}$ is implemented by the convolutional neural network architecture of the generator of DCGAN model (Radford et al., 2016) to map $Z^{(\ell)}$ into an approximated image $\hat{I}^{(\ell)}$. The $\mathbb{S}_n$-invariant pooler $p^{(\ell)}$ is a network operating on each small $h \times w$ grid graph to produce the corresponding node feature for the next level $\ell + 1$. MGVAE is trained to reconstruct all resolution images. Fig. 13 shows an example of reconstruction at each resolution on a test image of MNIST (after the network converged).

We evaluate our MGVAE architecture on the MNIST dataset (LeCun et al.) with 60,000 training examples and 10,000 testing examples. The original image size is $28 \times 28$. We pad zero pixels to get the image size of $2^5 \times 2^5$ (e.g., $H^{(5)} = W^{(5)} = 32$). Each cluster is a small grid graph of size $2 \times 2$ (e.g., $h = w = 2$). Accordingly, the image sizes for all resolutions are $32 \times 32$, $16 \times 16$, $8 \times 8$, etc. In this case, the whole network architecture is a 2-dimensional quadtree. The latent size $d_z$ is selected as 256. We train our model for 256 epochs by Adam optimizer (Kingma & Ba, 2015) with the initial learning rate $10^{-3}$. In the testing process, for the $\ell$-th resolution, we sample a random vector of size $d_z$ from prior $\mathcal{N}(0, 1)$ and use the decoder $d^{(\ell)}$ to decode the corresponding image. We generate 10,000 examples for each resolution. We compute the Frechet Inception Distance (FID) proposed by (Heusel et al., 2017) between the testing set and the generated set as the metric to evaluate the quality

| Method | FID$_\downarrow$ ($32 \times 32$) | FID$_\downarrow$ ($16 \times 16$) | FID$_\downarrow$ ($8 \times 8$) |
|---|---|---|---|
| DCGAN | 113.129 | | |
| VEEGAN | 68.749 | N/A | N/A |
| PACGAN | 58.535 | | |
| PresGAN | 42.019 | | |
| **MGVAE** | **39.474** | 64.289 | 39.038 |

Table 12: Quantitative evaluation of the generated set by FID metric for each resolution level on MNIST. It is important to note that the generation for each resolution is done separately: for the $\ell$-th resolution, we sample a random vector of size $d_z = 256$ from $\mathcal{N}(0, 1)$, and use the global decoder $d^{(\ell)}$ to decode into the corresponding image size. The baselines are taken from (Dieng et al., 2019).

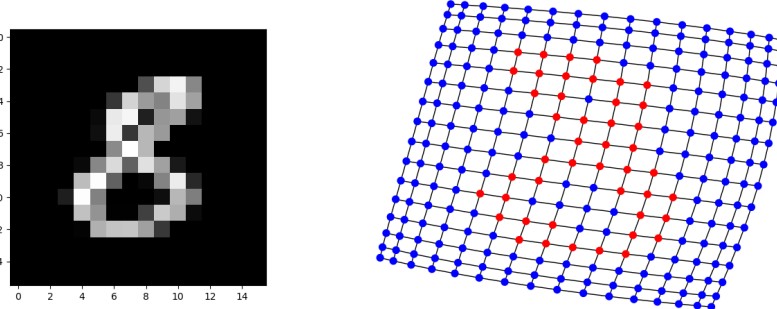

Figure 12: An image of digit 8 from MNIST (left) and its grid graph representation at $16 \times 16$ resolution level (right).

of our generated examples. We use the FID implementation from (Seitzer, 2020). We compare our MGVAE against variants of Generative Adversarial Networks (GANs) (Goodfellow et al., 2014) including DCGAN (Radford et al., 2016), VEEGAN (Srivastava et al., 2017), PacGAN (Lin et al., 2018), and PresGAN (Dieng et al., 2019). Table 12 shows our quantitative results in comparison with other competing generative models. The baseline results are taken from *Prescribed Generative Adversarial Networks* paper (Dieng et al., 2019). MGVAE outperforms all the baselines for the highest resolution generation. Figs. 14 and 15 show some generated examples of the $32 \times 32$ and $16 \times 16$ resolution, respectively.

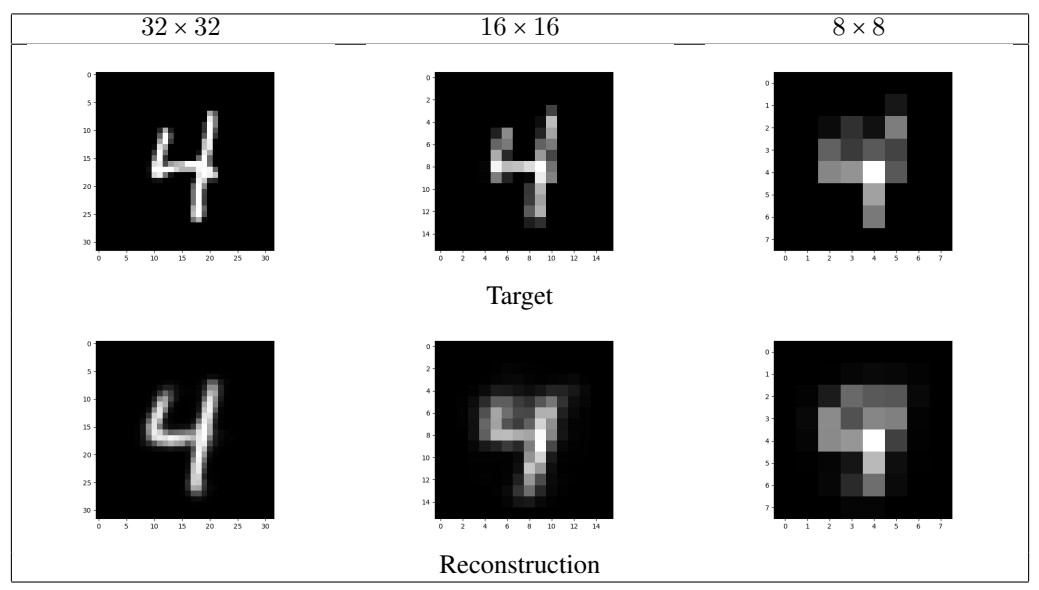

Figure 13: An example of reconstruction on each resolution level for a test image in MNIST.

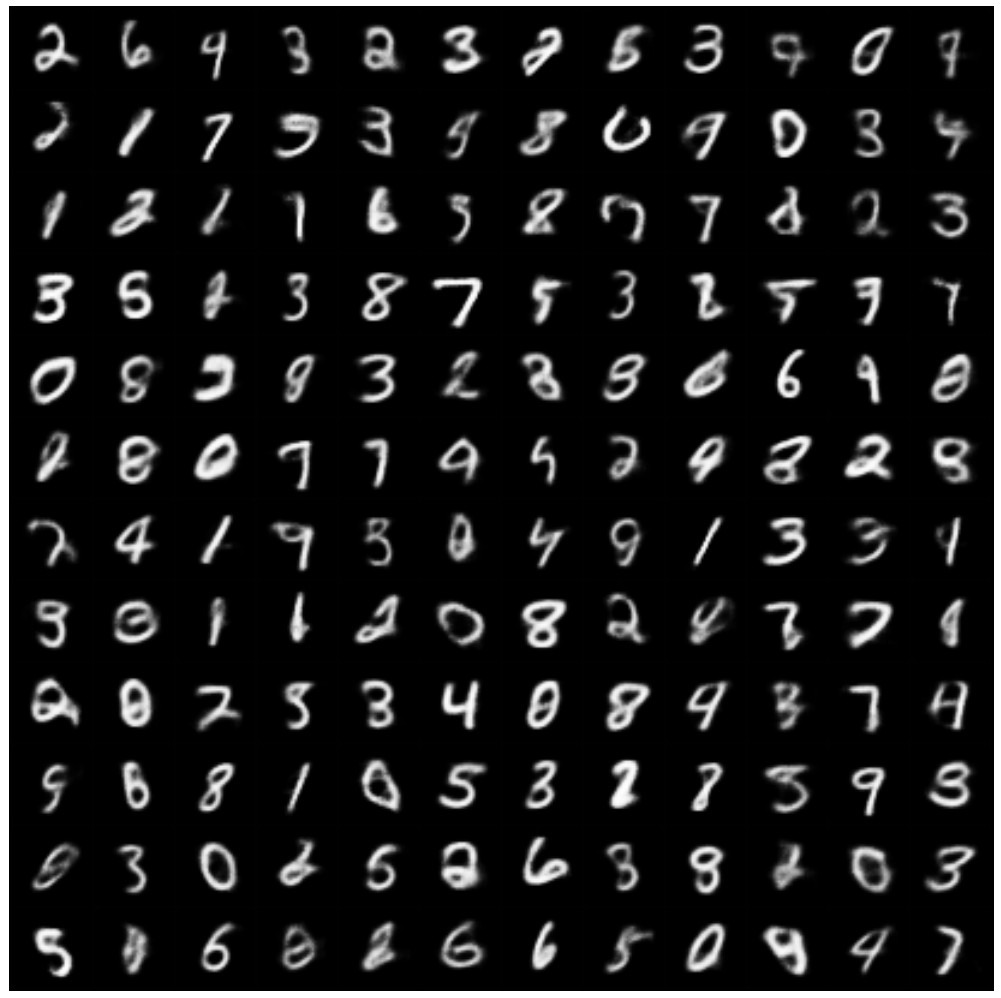

Figure 14: Generated examples at the highest $32 \times 32$ resolution level.

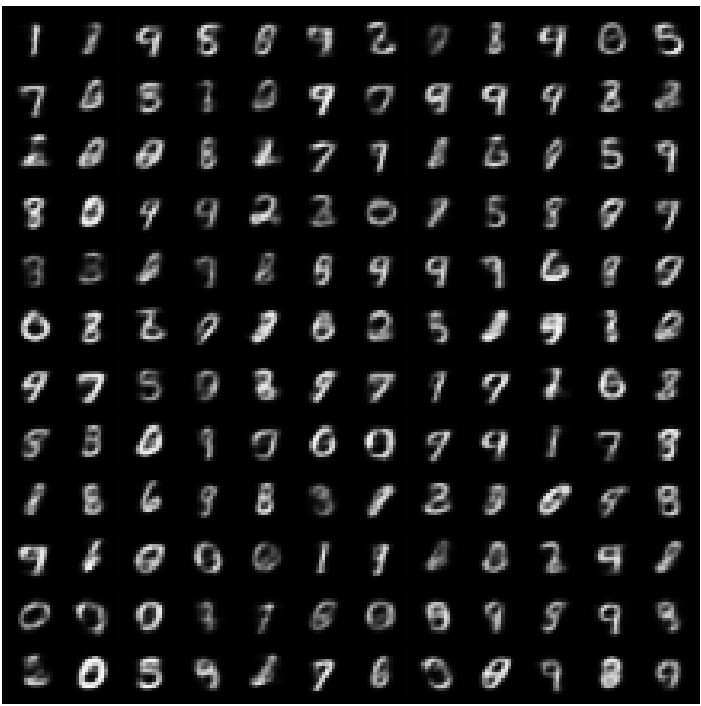

Figure 15: Generated examples at the $16 \times 16$ resolution level.

