# OpenReview forum: "Multiresolution Equivariant Graph Variational Autoencoder"
_ICLR.cc/2022/Conference — ICLR 2022 Submitted_

### Official Review · Reviewer_XesM · 2021-10-26

**Correctness:** 3
**Technical Novelty And Significance:** 2
**Empirical Novelty And Significance:** 2
**Recommendation:** 5
**Confidence:** 4

**Details Of Ethics Concerns:**

Nil

**Main Review:**

The paper is quite technical. Although the paper touches a number of issues regarding the equivariant and invariant concepts, this comes from the basic requirement from the GNN, see Michael Bronstein et al 5G paper. That is, the main contents in Section 3 can be regarded the major contribution of the paper. In other words, the main contribution comes from Section 4 where a hierarchical type of VAE framework is proposed, although building on the existing approaches or techniques.

The paper is readable in its current format. I dont see any specific theoretical issues and the mathematical derivation is solid (anyway this is the standard steps of building VAE models). It is nice for the authors to deal with the non-differentiable issue of the clustering assignment, i.e., eqn (1). The application of the Gumbel-max trick makes BP training possible.  However I am a bit confused or surprised that the authors did not mention at all on how to deal with the differentiable issue with the cardinality function in the KL term, see eqn (2) or (3).  It seems to me this term is non-differentiable due to counting the number of nodes in clustering which is determined by the model parameters in a complicated way.  Or the authors may make their codes available to the reviewers at this stage just like other authors do in this round.



**Summary Of The Paper:**

This paper proposes Multiresolution Equivariant Graph Variational Autoen-coders (MGVAE) which can learn and generate graphs in a multiresolution and equivariant fashion. Higher order message passing is used to encode the graph while maintaining learning mutually exclusive clusters so that coarsening into a lower resolution, thus creating a hierarchy of latent distributions. The model also maintain an end-to-end permutation equivariant with respect to node ordering. Their experimental results show that MGVAE achieves competitive re-
sults with several generative tasks and graph link prediction etc.

**Summary Of The Review:**

My bit concern is the way of how to handle the KL terms in the objective.

---

> ### Author Response · Authors · 2021-11-15
> **Permutation equivariance & Differentiablity**
>
> Thanks for your review!
>
> Equivariant latent & generative model:
> (1) Yes, it is true that permutation equivariance and invariance is the basic requirement for GNN. However, the traditional MPNN achieves this requirement by summing up the information coming from each node's neighborhood, but this also limits the representative power of the model. For example, let's consider the sum of 2 vectors (0, 1) + (1, 0) = (0, 0) + (1, 1). Both cases have the same result (1, 1). Obviously, the summation cannot distinguish between the two cases. Therefore, we employ the second-order message passing as in (Kondor et al., 2018) and (Maron et al., 2019) to increase the model's representative power.
>
> (2) The second-order networks are particularly essential for us to extend the original variational autoencoder (VAE) (Kingma & Welling, 2014) model that approximates the posterior distribution by an isotropic Gaussian distribution with a diagonal covariance matrix and uses a fixed prior distribution N(0,1). We generalize by modeling the posterior by N(\mu, \Sigma) in which \Sigma is a full covariance matrix, and we learn an adaptive parameterized prior instead of a fixed one. Only the second-order encoders can output a permutation equivariant full covariance matrix, while lower-order networks such as MPNNs are unable to.
>
> Gumbel-max trick & KL term:
> (1) By using the Gumbel-max trick (builtin in PyTorch), we can have a hard assignment matrix in which each row corresponds to a node, each column corresponds to a cluster and exactly one element in each row is 1 while the rest is 0.
> (2) The cardinality of each cluster can be computed easily (in PyTorch) by summing each row of the assignment matrix. Then, the balanced-cut loss is implemented as the KL divergence between the (normalized) clusters' cardinalities vs. the uniform distribution. It is exactly as mentioned in Equation 2 and 3 in the main text. Every operation is simple, differentiable, and builtin already in PyTorch.
>
> I hope that my response answered your concern. Please let us know!

---

> > ### Comment · Reviewer_XesM · 2021-11-24
> > **Thanks**
> >
> > Thanks for the answer "The cardinality of each cluster can be computed easily (in PyTorch) by summing each row of the assignment matrix."     Then if this is the case, please write |V| in terms of \Pi.

---

### Official Review · Reviewer_dMFu · 2021-10-28

**Correctness:** 2
**Technical Novelty And Significance:** 2
**Empirical Novelty And Significance:** Not applicable
**Recommendation:** 5
**Confidence:** 4

**Main Review:**


**Strengths**:
* Nice introduction.
* Figure 1 is a nice illustration of def 1 and 2.
* Overall clearly written.
* Good motivation for why multi-resolution graph generation can be useful.

**weaknesses**:
* At the end of section 3.1, the authors claim that MGN (multiresolution graph networks) "is more efficient than existing methods in the field." The rational is that the prediction of the adjacency matrix in the decoder happens hierarchically, and at each stage only local adjacency need to be predicted, as opposed to the full adjacency matrix prediction in one-stage decoders. However, empirical results seem to contradict this claim of efficiency. In Section 5.1 on molecule generation, the authors state the following "because of high complexity, we only train on a small random subset of examples while all other methods are trained on the full datasets. "  This contradiction with the earlier claim needs to be clarified. What is the source for the high complexity that prevents this method from being trained on the entire dataset (which is not that large)? Without any explanation on what the source of the complexity is, one cannot estimate if this method is scalable to larger graphs.
* Although there are quite a few experiments in the paper, they don't give sufficient insight into why the proposed method works better and which contributions are important. For example, the influence of the multi-resolution part of the generation and encoding is not investigated with an ablation study, whereas this is one of the main components of the paper. I would expect to see ablation studies that study the influence of the number of stages in the multi-resolution generation and encoding, as well as the influence of learnable balanced partitions versus fixed graph partitioning. Furthermore, what is the effect of the learnable equivariant prior over the standard normal prior?
* The definitions in section 3 take up unnecessarily much space and distract from the contributions of the paper. Example: def. 5 and 6 are used throughout the field and don't require that much space. Definition 9 again has much overlap with 5 and 6. Although writing a didactic paper with sufficient background is important, in this case I think it is a bit overdone, and more space and discussion should be spent in the main paper on experimental results and details of the implementation and method. For example, the matching scheme for the learnable prior with full covariance matrix will have an influence on the complexity of the method but gets almost no attention in the main paper.

**Minor comments/questions**:
* What does "variationally decode" mean in the abstract. VAE's use the encoder to do variational inference, not the decoder.
* Bottom of page 3: refers to figure 3.1 but should probably be figure 2.
* In the paper symbol $\mathcal{N}$ is used both for gaussian probability distribution and a neural network in definition 9. It would improve clarity if there are separate symbol for these two things.
* The decoder distribution at the bottom of page 6 only considers the case of unweighted graphs for which the adjacency matrix contains elements of value 0 or 1, but not the weighted case. Could you extend it to the weighted case?
* What does the index k mean in the last sentence of the first paragraph in section 4.2 in the equation for $\mathcal{Z}^l_i$? The LHS doesn't depend on k, but the RHS does.
* Appendix C, typo: "preserves equiarience" --> "preserves equivariance".


**Summary Of The Paper:**

This submission considers a multiresolution graph auto-encoder framework which is equivariant with respect to node permutation. Contrary to prior work on multi-scale graph generation, in this work the hierarchical structure in the encoder is learned through differentiable graph coarsening and is encouraged to produce balanced partitions of the graph. In the decoder, the graph is generated hierarchically and at each level, local adjacency matrices need to be predicted, rather than directly predicting the adjacency matrix of the full graph.  The method is evaluated on molecular generation, community graph generation,  and citation network generation in the main paper. The supplementary materials contains additional experiments on link prediction, unsupervised and supervised molecular property predictions, and graph-based image generation.


**Summary Of The Review:**

Although I think this paper has potential, in its current form I vote to reject it. The weaknesses raised in the previous part of the review are the most important reason for this vote. Although each one of them consists of a significant weakness from my point of view, they are listed in order of descending importance.

---

> ### Author Response · Authors · 2021-11-26
> **On complexity and learnable equivariant prior**
>
> Thanks for your review!
>
> Thanks for raising questions regarding the complexity. As we mentioned in section 3.1, the multiresolution graph networks (MGN) is more efficient than other methods. The reason is: we employ local encoders and local decoders instead of global ones. Each local network operates on a subgraph or cluster only, instead of the whole graph. We want to emphasize that MGN is a generic framework in which the actual implementation of encoders and decoders can be customized. If the encoders are just message passing neural networks (MPNN) or simple graph neural networks, and the decoders are just simple inner product, then the complexity is actually lower than using MPNN alone on the whole graph. The high complexity that we mentioned in section 5.1 as you pointed out is from the use of second-order networks  (Kondor et al., 2018) (Maron et al., 2019). The second-order networks have higher complexity comparing to MPNN. But again, our proposed MGN framework can work with MPNN too and in the case we exploit the sparse implementation of MPNN then the framework is very efficient.
>
> It is essential to have the learnable (parameterized) Gaussian prior with full covariance matrix for the higher-order latent to work. The standard normal prior is indeed not enough. Here is our normal-word explanation. First, let's suppose the latent is represented by a vector R^d with d is the number of channels and is independent of the number of nodes N. Obviously, the original VAE with the standard normal prior was designed for such a latent. But the problem is: we want to decode this R^d latent into the adjacency matrix in R^{N x N}. And there is no permutation equivariant way to do so. Thus, we must have the latent represented by a matrix R^{N x d} in which each row of this latent corresponds to a node. Now, there exists permutation equivariant mapping to decode this N x d latent into a N x N matrix, for example simple inner product (we proposed the use of second-order decoder). However, if we use the isotropic Gaussian prior as in the original VAE, another problem happens: all nodes are independent of each other!! That is not what we want, because in the generation process, we have to sample from our prior, and if the nodes are independent then the decoded topology (adjacency) would certainly violate chemical constraints (e.g., validity, valence, etc.). We want all the nodes/atoms are correlated. Therefore, we must use the parameterized Gaussian prior with full covariance matrix. And the interpretation of this new model is learning Markov Random Fields as we detailed in the appendix.
>
> I hope that this answers your question. Thanks for pointing out the typos also.

---

### Official Review · Reviewer_KkPB · 2021-10-30

**Correctness:** 4
**Technical Novelty And Significance:** 3
**Empirical Novelty And Significance:** 3
**Recommendation:** 5
**Confidence:** 4

**Main Review:**

This paper is well-motivated and easy to follow. I have some minor concerns below for the authors to address.

Minor concerns:

1. The literature review could be better. Recent studies have widely acknowledge the impacts of progressive or hierarchical graph generation, especially for molecular generation, e.g., *Hierarchical Generation of Molecular Graphs using Structural Motifs* and its following works. These works should also be the baselines for the proposed method.

2. In the *Learning to Cluster* section, in order to make the clustering procedure learnable, Gumbel-max trick is adopted to replace the max-pooling. Do you use Gumbel-softmax in practice since Gumbel-max is not differentiable. If the Gumbel-softmax is used, then whether the complexity of $O({|V|^2}/{K})$ could not be hold since Gumbel-softmax is not hard partition.

3. It's better to present empirical evidence to demonstrate the claim about the efficient of time and space complexity for the proposed framework.

4. I have a strong concern about the metrics used in graph generation tasks, although they're usually used in previous literatures. More concrete, in the molecular generation (Figure 3 and Table 1), while the chemical validity is 100% on both dataset, the generated molecular structure seems wired and are most likely not stable. For example, I have asked my colleagues who have chemical-related backgrounds, and they say that most molecules shown in Fig. 3 (at least 1st, 2nd, 3rd, 5th, 8th) are incapable of stable existence. Please kindly correct me if there is anything wrong due to my limited chemical knowledge.

5. Comparing the propsed framework with more recent baseline methods rather than only comparing to classical methods would make the experimental results more solid. I suggest that for the molecular generation and representation tasks, the authors could add some additional experiments to comparing baseline methods published in recent years, and especially the state-of-the-art method (https://paperswithcode.com/sota/graph-regression-on-zinc-500k).

6. Capturing the graph structure is more essential for molecular modeling, I encourage the authors exploit more on molecular generation and representation, instead of image generation or social netoworks.

Presentation problems or typos:

1. Section 3.1 Definition 4 use $\mathbf{d}_{local}$ to refer to local decoder, Section 4.2 use $\mathbf{d}_{local}$ to refer to local graph encoder.

2. In section 3.1, subscripts $k$ of $\mathcal{Z}_k$ and $\mathcal{G}_i$ are clusters indices in a graph. In section 4.2, subscripts $i$ of $\mathcal{Z}_i$ and $\mathcal{G}_i$ are sample indices, notation is changed to $[\mathcal{Z}_{i}]_k$ and $[\mathcal{G}_{i}]_k$.

3. In section 4.2, there is a subscript of a set, which is confusing. May replace $Z_{i}^{(l)}=\{[Z_{i}^{(l)}]_k \in \R^{|[\mathcal{V}_i^{(l)}]_k|\times d_z}\}_k$ by $Z_{i}^{(l)}=\{[Z_{i}^{(l)}]_k \in \R^{|[\mathcal{V}_i^{(l)}]_k|\times d_z}| 1\leq k \leq K_{i}^{(l)}\}$.

   And $Z^{(l)}(i)=\{Z_{k}^{(l)}(i) \in R^{|\mathcal{V}_k^{(l)}(i)|\times d_z}| 1\leq k \leq K^{(l)}(i)\}$ may be better.

4. In appendix section A Definition 11, $j$ in the indices of C should be redundant(typo).


**Summary Of The Paper:**

This study presents a multi-scale graph VAE with hierarchical graph coarsening. The Gumbel-max trick is employed to generate learnable hard partition for clustering, and permutation equivariant tensor operations are used (Kondor et al., 2018) to construct the group equivariance network. The authos find that the proposed framework exhibits competitive performance in graph generation, molecular generation, molecular representation learning, link prediction and graph-based image generation. Given the applicability of this framework, this proposed framework may be interesting to the graph generation community, and this paper is well-written.

**Summary Of The Review:**

I'm definately willing to raise my score if my concerns have been well addressed.

---

> ### Author Response · Authors · 2021-11-26
> **A principled approach to graph generation**
>
> Thank you for your detailed review!
>
> 1. Regarding the hierarchical graph generation, thanks for pointing out "Hierarchical Generation of Molecular Graphs using Structural Motifs" (Jin et al., 2020) that I think is a relevant paper. However, (Jin et al., 2020) is still not an equivariant model. Theoretically saying, given a non-equivariant model, for an input adjacency matrix A of size N x N and its corresponding reconstructed (by the decoder) adjacency B, the reconstruction loss function that compares A and B must consider all N! (factorial) possibilities of matching. This is indeed an NP-hard problem. You can, in theory, define the reconstruction loss by solving a quadratic assignment problem to match A to B, but this is intractable for large N. For small molecules, people in the field just sort the set of atoms in a particular order by some heuristics. Our work is a more principled approach to the graph generation problem. Here, we propose end-to-end equivariant generative model, that means if we permute/transform A then the reconstructed B would transform accordingly, and we can avoid the above issue of non-equivariant methods including (Jin et al., 2020). Furthermore, (Jin et al., 2020) extracts the motifs (substructures) explicitly in a separate process by a heuristics algorithm. In contrast, our work learns to extract the substructures in a data-driven manner (i.e. learning to cluster). In addition, (Jin et al., 2020) is designed only for molecular generation, while our generative model can be applied to a wide range of problems including general graph generation.
>
> 2. Regarding the learning to cluster algorithm and the Gumbel-softmax, you can get the hard assignment (binary 0/1) out with the PyTorch implementation of Gumbel-softmax https://pytorch.org/docs/stable/generated/torch.nn.functional.gumbel_softmax.html and this operator is differentiable. I think this should address your concern.
>
> 2 and 3. That complexity still holds. As the link to the PyTorch's Gumbel softmax operator above, we can get the hard assignment. We want to emphasize that multiresolution graph network (MGN) is a generic framework that employs local encoders and local decoders. The actual implementation of these local networks can be customized. For example, the encoders can be MPNN or simple graph nets, and the decoders are simple inner product, this case the whole framework is very efficient because local networks only operate on subgraphs (clusters) instead of the whole graph. The high complexity only comes from the use of second-order equivariant models (Kondor et al., 2018) (Maron et al., 2019) for these encoders/decoders.
>
> 4. Regarding the metrics for molecular generation, I think what the field is missing is a common benchmark. We include more statistics of drug-likeness in the appendix along with more example generated molecules. I hope you can more stable and drug-like molecules in the appendix.
>
> 5. Thanks for pointing out the supervised graph regression benchmark for ZINC 500k. In the appendix, we evaluated our multiresolution graph network (MGN) in a smaller dataset ZINC 10k, and compared to published methods until year 2020. The main contribution of our paper is the multiresolution and equivariant generative model MGVAE, while MGN is a minor contribution.
>
> 6. I think our work is a more principled way to look at generation problems in general: multiscale/multiresolution for large graphs like citation graphs and for images is interesting and will certainly have useful applications in the future.
>
> Hope that my replies answer your concerns. Thanks for pointing out the typos also!

---

### Official Review · Reviewer_867y · 2021-11-02

**Correctness:** 4
**Technical Novelty And Significance:** 3
**Empirical Novelty And Significance:** 1
**Recommendation:** 5
**Confidence:** 3

**Main Review:**

Strengths

- The proposed method is presented in a very clear and structured way.

- The approach is well-motivated, adapting and integrating aspects of various other work (VAEs, learnable hard clusterings, balanced cuts, etc.) in a seamless manner.

- By being designed to be inherently permutation-equivariant with respect to node ordering, the MGVAEs lend themselves well for graph encoding and generation in a multiscale fashion.

- The proposed method constitutes a novel approach towards generating graphs.

Weaknesses

- The work by Jin et al. (ICML 2020) seems highly relevant in the context of the proposed method. Similar to the proposed approach, they adapt VAEs for the task of hierarchical predictions of molecular graphs in a coarse-to-fine fashion. While there are of course clear differences between the approaches, a discussion of and comparison with that work seems essential for the proposed method.

- In the abstract, the authors claim competitive results in graph generation, molecular generation, unsupervised molecular representation learning, link prediction on citation graphs, and graph-based image generation. However, the evaluation of the proposed method seems limited, especially in the main paper. For example, for the task of molecular graph generation, only validity, novelty, and uniqueness of the generated molecules are evaluated. Why are no additional metrics from standardised benchmarks for molecular generation evaluated (cf. Polykovskiy et al., 2018)?
Further, other work that has scored higher on the Citeseer dataset has been left out of the comparison (e.g., Davidson et al. 2018). The way in which the results are presented suggest that no better methods exist. I can understand if the goal of the results is to merely highlight competitive performance with respect to ‘popular’ methods and not to claim state of the art performance. However, putting the results into context with respect to the current state of the art still seems important to gauge the presented method properly.

- The authors extend the prior of the VAEs from an isotropic Gaussian to a parameterized Gaussian with learnable mean and covariance. While this constitutes an interesting extension, the importance for the task at hand is unclear to me. What effect does this have on the learnability of the tasks or on performance?

- The authors highlight that the approach allows to generate graphs at various levels of resolution. Under which circumstances might this be useful? What is the meaning of a low-resolution molecule?




**Summary Of The Paper:**

The authors present a variational-autoencoder-based model for learning and generating graph structures. In particular, they propose a multiresolution graph network (MGN) that encodes a given graph in a hierarchical manner, i.e., at different levels of resolution.
Training the nodes of the coarsened graphs as latents of a variational auto-encoder allows for sampling a new graph in a bottom-up manner, i.e., at increasingly higher resolution.

The main contribution of this work consists of developing the framework of MGNs and demonstrating that they can be trained as hierarchical variational autoencoders, which yields the multiresolution graph variational autoencoders (MG-VAEs). The MG-VAEs represent generative models that allow for generating graphs in a multiresolution and equivariant manner.
The authors evaluate these models in a range of different settings, from unsupervised representation learning, over link prediction on citation graphs, to the generation of molecules or graph-based image generation.


**Summary Of The Review:**

The authors present a novel approach for generating complex graphs, e.g., molecules by introducing a model that allows for sampling from a latent space at various levels of resolution.
Further, the paper is well written and generally presented in a clear manner. However, the evaluation of the method and the comparison to other work should be expanded upon. In particular, relevant work seems to be missing both for providing context of the work in the current state of the field, as well as in the evaluation and comparison of the model to other work.
Thus, while I find the method worthy of acceptance at  ICLR, I deem the paper marginally below the acceptance threshold in its current state.

Edit:
The authors clarified my question regarding the parameterized Gaussian and I appreciate their time for doing so. I would encourage the authors to include this motivation for their approach also in the manuscript (while the method is formally described in the main text and the appendix, I cannot find a discussion for why this approach is taken). This will help readers understand the importance of this aspect better.

Unfortunately, my main concerns regarding comparison with and discussion of relevant related work remain unanswered and I thus maintain the score of 5.

---

> ### Author Response · Authors · 2021-11-26
> **Regarding parameterized Gaussian prior and low-resolution molecule**
>
> Thanks for your review!
>
> It is essential to have the parameterized Gaussian prior with full covariance matrix for the higher-order latent to work. Here is a normal-word explanation (while we have discussed formally in the main text and section B, C of our appendix). First, let's suppose the latent is represented by a vector R^d with d is the number of channels and is independent of the number of nodes N. Obviously, the original VAE was designed for such a latent. But the problem is: we want to decode this R^d latent into the adjacency matrix in R^{N x N}. And there is no permutation equivariant way to do so. Thus, we must have the latent represented by a matrix R^{N x d} in which each row of this latent corresponds to a node. Now, there exists permutation equivariant mapping to decode this N x d latent into a N x N matrix, for example simple inner product (we proposed the use of second-order decoder). However, if we use the isotropic Gaussian prior as in the original VAE, another problem happens: all nodes are independent of each other!! That is not what we want, because in the generation process, we have to sample from our prior, and if the nodes are independent then the decoded topology (adjacency) would certainly violate chemical constraints (e.g., validity, valence, etc.). We want all the nodes/atoms are correlated. Therefore, we must use the parameterized Gaussian prior with full covariance matrix. And the interpretation of this new model is learning Markov Random Fields as we detailed in the appendix.
>
> Regarding the meaning of low-resolution molecules, one resolution level above the input molecule is indeed a tree-like structure, something similar to molecular backbone or skeleton. Indeed, our multiresolution VAE allows us to generate not only the molecules itself but also the molecular skeleton.
>
> Hope this explanation answers your questions.

---

### Official Review · Reviewer_PA3h · 2021-11-02

**Correctness:** 2
**Technical Novelty And Significance:** 3
**Empirical Novelty And Significance:** 2
**Recommendation:** 3
**Confidence:** 4

**Main Review:**

Strength:
The proposed method is presented with many details. Several learning tasks are designed to support the novelty of the proposed method;

Weakness:
1. The main focus of the proposed method is its equivariance property. It would be nice to give some counter-examples that existing methods fail to guarantee this condition, which then becomes problematic.
2. A main concern of the proposed method is its "high complexity". The first experiment only supports the training "on a small random subset of examples", according to the authors. Although the results in Table 2 tries to prove the complexity issue is minor, more evidence (probably more scenarios or perspectives?) could be provided to persuade. A similar issue arises in the second experiment, where the authors select a subset of two considerably small graphs, Cora and Citeseer.

Minor issues:
1. The concepts should be first introduced BEFORE further discussion, e.g, equivariance.
2. Many details should be carefully looked after. For example, the in-text citations are disordered. Section 1 "...between their discrete substructures (subgraphs) (You et al., 2018a) (Li et al., 2018) (Liao et al., 2019)...", the names should be put in one parenthesis.
3. The reproducibility statement was not addressed.
4. The presentation should be taken care of. For example, "...we generalize by modeling..." (page 5), what is generalized here? “On the another hand” (page 6), it sounds supernatural that one has 'another' hand.

**Summary Of The Paper:**

The authors propose a multi-resolutional graph generation tool with a variational autoencoder. The graph hierarchy is learnable and flexible to new data.

**Summary Of The Review:**

The paper could be more persuasive if it provides further justification regarding the necessity (and initativity) of designing an equivariance graph generation. The concern on model complexity should be addressed. Also, the paper should be carefully proofread before it is ready for acceptance.

---

> ### Author Response · Authors · 2021-11-14
> **Our work shed a new light into the graph generation problem**
>
> Thanks for your review!
>
> Our paper shed a new light to the graph generation problem. It is the first model that embodies both multiresolution and equivariance. For your question regarding an example of equivariance, it is trivial that every autogressive generative model breaks this important property. I believe we already mentioned about the significant difference between equivariant and autoregressive in the main text of our work. For autoregressive models, you always have to define before-hand a particular order of nodes to decode into the output graph given the latent. Basically, autoregressive models construct the graph sequentially by adding new node/edge one-by-one. But for a graph of size N, obviously you would have N! possibilities of doing so. Let say for a small molecule of 9 heavy atoms as in QM9, the number of ways to decode into exactly the same molecule (given a latent vector) is 362,880. In contrast, our proposed model is completely equivariant end-to-end that means you can permute the input graph arbitrarily, the latent structure would transform accordingly and so the reconstructed output graph.
>
> For your comment about high complexity, the main cost is from the second-order message passing. But it is important to note that our multiresolution and equivariant framework can work with lower-order message passing such as MPNN or simple GNN that has a much faster implementation.
>
> Finally, please justify your decision of giving us a score 3. Your comments about the weaknesses such as an example and high complexity are not convincing enough for such as a low score. The objective of our work is to shed a new light into the graph generation problem, and it is not just an engineering purpose. I hope that my answers to your comments are convincing and reasonable.

---

### Official Review · Reviewer_9mH3 · 2021-11-08

**Correctness:** 3
**Technical Novelty And Significance:** 3
**Empirical Novelty And Significance:** 3
**Recommendation:** 6
**Confidence:** 3

**Main Review:**

Strengths:

+ MGVAE is the first hierarchical generative model to learn and generate graphs both in a multiresolution and in an equivariant manner.
+ Lots of experimental results.


Weaknesses:

The paper needs a more compelling motivation for why the authors' specific approach to  generating graphs in a multiresolution and in an equivariant  manner is important, or why is it important to employ VAE, given that the (i) the idea of studying the properties of graphs via the eigenvalues and eigenvectors of their associated graph matrices at different resolutions is not novel, and neither is the (ii) permutation equivariance of graph neural networks .

The authors have asserted  that it is desirable to have (1) balanced K-cluster partition, (2)  local encoders which the authors have asserted tend to generalize better for same-size subgraphs and (3) uniform distribution of nodes into clusters. This maybe something observed by the authors in their specific application, but it is not generally true.

In imaging, nodes( pixels) are assigned adaptatively to clusters based on high gradients or high curvatures which results in clusters of different sizes and at different locations.

Why not perform an adaptive clustering by employing graph downsampling, graph reduction, and filtering and interpolation of signals on graphs?



Vasilescu20 and Vasilescu19 introduced a multiresolution hierarchical tensor factorization that model the mechanism that generated the data formation and employs a hierarchy of latent variables where the nodes (pixels) can be assigned to clusters based on adaptive strategy. They applied this approach to face recognition (Vasilescu02) and developed a generative multiresolution hierarchical TensorFaces model (Vasilecu19, Vasilescu20).


@inproceedings{Vasilescu20,

author={Vasilescu, M. Alex O. and Kim, Eric and Zeng, Xiao S.},
booktitle={2020 25th International Conference of Pattern Recognition (ICPR 2020)},
title={Causal{X}: {C}ausal e{X}planations and {B}lock {M}ultilinear {F}actor {A}nalysis},
year={2021},
location={Milan, Italy},
month={Jan},
pages={10736--10743}
}

@misc{Vasilescu19,

author={Vasilescu, M. Alex O. and Kim, Eric},
booktitle={The 25th ACM SIGKDD Conf. on Knowledge Discovery and Data Mining (KDD’19): Tensor Methods for Emerging Data Science Challenges Workshop},
title={Compositional Hierarchical Tensor Factorization: Representing Hierarchical Intrinsic and Extrinsic Causal Factors},
year={2019},
location={Anchrorage,AK},
month={Aug. 5},
}


@inproceedings{Vasilescu02,

  author =	 "M. A. O. Vasilescu and D. Terzopoulos",
  fullauthor =	 "M. Alex O. Vasilescu and Demetri Terzopoulos",
  title =	 "Multilinear Analysis  of Image Ensembles: {T}ensor{F}aces",
  booktitle =	 "Proc. European Conf. on Computer Vision (ECCV 2002)",
  address =	 "Copenhagen, Denmark",
  month =	 "May",
  year =	 "2002",
  pages =	 "447-460"
}

**Summary Of The Paper:**

The authors propose Multiresolution Graph Networks (MGN) and Multiresolution Graph Variational Autoencoders (MGVAE) to learn and generate graphs in a multiresolution and equivariant manner that has been applied to citation data, molecular data and  MNIST imaging data.

**Summary Of The Review:**

MGVAE is the first hierarchical generative model to learn and generate graphs both in a multiresolution and in an equivariant manner.  The experiments are extensive.  The paper needs a stronger motivation for why the authors' specific approach is important considering  that (i) the idea of studying the properties of graphs via the eigenvalues and eigenvectors of their associated graph matrices at different resolutions is not novel, and neither is the (ii) permutation equivariance of graph neural networks

---

> ### Author Response · Authors · 2021-11-14
> **We are confused by one of your claims**
>
> Thanks for your review!
>
> We did not propose any use of eigenvalues and eigenvectors. So, I am very confused when you said “ the idea of studying the properties of graphs via the eigenvalues and eigenvectors of their associated graph matrices at different resolutions is not novel”. Could you please explain? I suspect you might have misunderstood something from our proposal.
>
> You suggested the use of an adaptive clustering by employing graph downsampling, graph reduction, and filtering and interpolation of signals on graphs. To my best knowledge, adaptive clustering is a form of learnable clustering that uses Q-learning. Our proposal is also learnable clustering in which a graph neural network decides the assignment matrix at each level of resolution via back-propagation. I am not sure what the advantage of adaptive clustering really is in comparison with our learnable clustering algorithm, since both are learnable in a data-driven manner. You also mentioned graph downsampling and graph reduction. I think we have made it very clear in the main text that the graph size reduces after every resolution. Isn’t it downsampling and reduction? You also mentioned filtering. Indeed, the graph neural network is already built upon the message passing framework that generalizes the convolution operation to graph, which in some sense also acts as graph filtering. In conclusion, it seems to me that we already had all your suggestions in our work!

---

### Note · Authors · 2023-08-25
**Submission Withdrawn by the Authors**

I have read and agree with the venue's withdrawal policy on behalf of myself and my co-authors.

---

### Decision · Program_Chairs · 2022-01-20

**Decision:**

Reject

**Comment:**

The paper proposes multiresolution and equivariant generative models.  Experimental results for several applications are shown.

Pros:
- A first hierarchical generative model with multiresolution and equivariance.
- Extensive experiments

Cons:
- Marginal novelty (multiresolution and permutation equivalence each is not novel for graph neural networks.
- State-of-the-art methods are not compared as baselines.
- Some standard metrics are not evaluated, and the used metrics are questionable (some generated molecules might not be stable although the chemical validity is 100%).
- Time/space complexity evaluation is missing.

The authors did not address some of the serious concerns in the rebuttal.